# Loss of wall stress homeostasis in ascending thoracic aortic aneurysm: histomorphometric insights into patient variants

Berta H. Ganizada[1,2,3] , Koen D. Reesink[3,4], Shaiv Parikh[3,4], Pepijn J. M. H. Saraber[2,3,4],
Jack P. Cleutjens[3,5] , Austin Isabella[2,3] , Mathilde Q. E. Krebbekx[1,3], Armand M. Jaminon[2,3],
Cecile F. M. Maassen[2,3], Mitch J. F. G. Ramaekers[3,6,7] , Bart Spronck[3,4] , Koen van der Laan[3,4],
Rory R. Koenen[2,3], Roberto Lorusso[1,3], Jos G. Maessen[1,3], Elham Bidar[1,3] and Leon J. Schurgers[2,3]

[1] *Department of Cardiothoracic Surgery, Maastricht University Medical Centre, Maastricht, The Netherlands*
[2] *Department of Biochemistry, Maastricht University, Maastricht, The Netherlands*
[3] *Heart & Vascular Center (HVC), Cardiovascular Research Institute Maastricht (CARIM), Maastricht University Medical Center, Maastricht, The Netherlands*
[4] *Department of Biomedical Engineering, Maastricht University Medical Centre, Maastricht, The Netherlands*
[5] *Department of Pathology, Maastricht University Medical Centre, Maastricht, The Netherlands*
[6] *Department of Radiology and Nuclear Medicine, Maastricht University Medical Centre, Maastricht, The Netherlands*
[7] *Department of Cardiology, Maastricht University Medical Centre, Maastricht, The Netherlands*

Handling Editors: Harold Schultz & Nikki Jernigan

The peer review history is available in the Supporting Information section of this article (https://doi.org/10.1113/JP288734#support-information-section).

**Berta Ganizada** (Maastricht University), is a doctoral candidate in the Department of Cardiothoracic Surgery and Biochemistry at Maastricht University. Her research focuses on the histomorphometric analysis of cellular and extracellular matrix components in ascending thoracic aortic aneurysms, aiming to identify novel aortic wall morphometric markers that could guide surgical decision-making. This work is supported by the MAPEX research biobank, developed at the Cardiovascular Research Institute Maastricht (CARIM), which provides a structured platform for collecting and analysing human aortic tissues. Her research was honoured with the Young Investigator Lecture Award at the Artery Congress in Nancy, France, in 2022.

Ethical approval: The study was approved by the Ethics Committee of the Maastricht University Medical Centre with the number METC-2019-1235, adhering to the following criteria: Declaration of Helsinki and written consent for the use of body materials for study and research purposes.

**Abstract figure legend** Dynamic interplay between the extracellular matrix (ECM) and vascular smooth muscle cells (VSMCs) in the medial layer of ascending thoracic aortic aneurysms (ATAA) associated with two variants: thickening (hypertrophic) and thinning (hypotrophic) aneurysm remodelling.

**Abstract** Histomorphometric differences in cell-matrix properties were analysed between ascending thoracic aortic aneurysm (ATAA), dissection (ATAAD) and non-aneurysmal patients, as well as across the circumference of the aneurysms in ATAA cases. Fresh anterior aortic wall samples were collected during surgery. A significant radius-to-intima-media thickness (IMT) ratio variation was observed among ATAA patients, indicating patient-specific adaptive responses. The radius-IMT ratio was significantly lower in ATAAD patients. The quantity and quality of elastin and the quantity of collagen were particularly reduced in ATAAD compared to ATAA and non-aneurysmal aortas. Matrix degradation was accompanied by an increase in the density of vascular smooth muscle cells (VSMCs), albeit with reduced expression of VSMC contractile markers (calponin and $\alpha$-smooth muscle actin ($\alpha$-SMA)). Concomitantly ATAA and ATAAD samples exhibited increased markers (matrix metalloproteinase (MMP)-2/9) of proteolysis. Based on radius-IMT ratios we roughly identified 'thickening' and 'thinning' (i.e. hypertrophic and hypotrophic) aneurysm variants to capture the substantial variation in the loss of mechanical homeostasis in ATAA. Interestingly we did not find conspicuous differences along the circumference of excised aneurysms in ATAA, except for an increased IMT heterogeneity in 'thinning' aneurysms. We conclude that during aneurysm formation wall stress homeostasis may remain partially intact, particularly in 'thickening' ATAA. Our study underscores the current critique that aneurysm dimensions are poor risk predictors; therefore there is a crucial need for better-informed preventive intervention in ATAAD.

(Received 13 February 2025; accepted after revision 10 June 2025; first published online 29 June 2025)

**Corresponding author** L. J. Schurgers: Cardiovascular Research Institute Maastricht, Department of Biochemistry, Maastricht University, Universiteitssingel 50, 6229 ER Maastricht, The Netherlands. Email: l.schurgers@maastrichtuniversity.nl

## Key points

- Ascending thoracic aortic aneurysm (ATAA) variants can be categorised as aortic medial thickening (hypertrophic) or aortic medial thinning (hypotrophic) based on the radius-to-intima-media thickness (IMT) ratio, reflecting distinct disruptions in mechanical homeostasis.
- Morphological patterns arise from dynamic interactions in the aortic medial layer between vascular smooth muscle cells (VSMCs) and the extracellular matrix (ECM).
- Increased number of synthetic VSMCs in the medial layer of ATAA patients is a compensatory response to maintain vessel elasticity and structural integrity.
- In acute type A aortic dissection (ATAAD) aortas with medial thinning are characterised by ECM breakdown and maladaptive remodelling.
- ATAA development is circumferentially homogeneous, despite the occurrence of inter- and intra-patient variability in vascular architecture, composition and VSMC characteristics.

## Introduction

Ascending thoracic aortic aneurysm (ATAA) is a precursor to aortic dissection (ATAAD), resulting from ascending aortic media degeneration (Isselbacher, 2005; Johansson et al., 1995; Pape et al., 2007; Perez et al., 2023; Vianna et al., 2022; Wang et al., 2021). Several clinical, and particularly genetic, risk factors, haemodynamic triggers and elastin-matrix degeneration are clear hallmarks of this disease spectrum (Isselbacher et al., 2022; Pape et al., 2007; Vianna et al., 2022). At a more fundamental level, however, the pathophysiological mechanisms compromising aortic tissue integrity and resilience are not yet fully understood. Particularly evidence regarding cell-matrix cross-talk in mechanical homeostasis, or its loss, remains lacking (Ganizada et al., 2023). Before introducing our study

we briefly review the current evidence and related perspectives.

An ATAA is typically considered to be a structural outward bulging of the aortic wall, referred to as outward remodelling. The bulging is associated with increased or heterogeneous fluid shear stress at the lumen-wall interface caused by aortic valve pathology and abnormal blood inflow angle (Kiema et al., 2022; Salmasi et al., 2023). In addition it is a well-established risk factor for aortic dissection (Erbel et al., 2014; Nienaber & Clough, 2015), supporting existing notions of blood pressure as a prime determinant of ATAAD development and events (Debeij et al., 2023). Genetic disorders, including Marfan syndrome, Loeys–Dietz syndrome, Ehlers–Danlos syndrome, Turner syndrome and EMILIN1 and MYH7/MYH11 mutations, are known to predispose to ATAAD (Nienaber & Clough, 2015).

The media is the functionally compliant and load-bearing layer of the aortic wall and comprises predominantly VSMCs and extracellular matrix (ECM), including major fibrous structures of elastin and collagen, forming a three-dimensional tubular structure of multiple concentric elastic lamella (Clark & Glagov, 1985; O'Connell et al., 2008). The ECM components interact with the VSMC to maintain – also on the long term – the elastic expansion and recoil of the aortic wall after each heartbeat/pulse wave. Medial matrix degradation due to ageing or disease eventually becomes irreversible, as signalled by increased aortic wall stiffness (Guzzardi et al., 2015; Yanagisawa & Wagenseil, 2020). Types I and III collagen constitute 90% of the collagen content in the aorta and exhibit age-related changes that increase ascending aortic stiffness (Cattell et al., 1996; Hosoda et al., 1984). Elastin – essential for aortic elastic expansion and recoil – is formed and structurally integrated in early childhood (Wagenseil & Mecham, 2007). When the quantity or quality of the elastin structure changes, aortic wall tissue structural integrity is clearly compromised (Yanagisawa & Wagenseil, 2020).

VSMCs may facilitate ECM degradation by secreting matrix metalloproteinases (MMPs), such as MMP-2, MMP-3, MMP-8, MMP-9 and MMP-13 (Chou et al., 2016). Tissue inhibitors of MMPs (TIMPs) normally regulate MMP activity (Khokha et al., 2013). Therefore an imbalance between MMPs and TIMPs is associated with pathological ECM turnover and alterations in cell behaviour (Baker et al., 2002). Aside from ECM turnover VSMCs also play a role in short- and long-term tissue responses and adaptation: blood vessels normally respond homoeostatically to changes in mechanical wall stress (Ganizada et al., 2023; Humphrey et al., 2015; Reesink & Spronck, 2019). For example in cases of hypertension the arterial walls thicken to counteract persistent increase in wall stress. In aneurysm development wall thickening would mitigate the development of excessive mechanical load (Humphrey et al., 2015). In a recent review we found that – contrary to popular notion – aortic dilation does not *per se* involve wall thinning (Debeij et al., 2023).

**Accordingly the research question for our mechanistic study is: to what extent is wall stress homeostasis absent or deranged in ATAAD?**

A comprehensive analysis of human tissue-based evidence at the cell-matrix level focused on the loss of mechanical homeostasis in ATAAD is currently lacking (Ganizada et al., 2023; Guzzardi et al., 2015). Additionally from our observations in clinical practice it is apparent that, while patients may arrive at a similar disease stage (i.e. preventive surgical ATAA repair in our cohort), rather diverse pathological manifestations occur at the tissue level (Ganizada et al., 2023). Therefore in our analysis we will carefully consider variability in our data. Previous histopathological studies established the role of medial ECM degradation in ATAAD (Buja et al., 2024; Grewal et al., 2021; Osada et al., 2018). However most human study designs were limited to comparing sporadic and genetic dissection cases and did not consider the more nuanced tissue remodelling and degradation observed in surgical ATAA repair cases.

In the present study we examine the histomorphometric and immunohistochemical characteristics of VSMCs and ECM in the media of residual tissues from elective ATAA and acute type-A dissection cases, as well as from control aortas obtained in patients undergoing coronary artery bypass grafting (CABG) or aortic valve replacement (AVR). By comparison of these groups our analysis and interpretation focus on understanding the loss of tissue integrity. In the ATAA group samples around the full circumference of the resected tissues allowed us to further scrutinise for signs of intact or dysfunctional wall stress homeostasis.

## Materials and Methods

### Ethical approval and study population

Study approval was obtained from the Ethics Review Board of Maastricht University Medical Centre (MUMC+), the Netherlands (METC No. 2019-1235) (Ganizada et al., 2023). We included consecutive patients who underwent surgical repair for thoracic aortic aneurysm (ATAA) ($n = 60$) or type A dissection (ATAAD) ($n = 17$). Patients with non-dilated aortas who underwent CABG or AVR without any history of aortic disease were included as controls ($n = 12$). Patients with

**Table 1. Study population baseline characteristics**

| Clinical data | Control CABG/AVR n = 12 | ATAA aneurysm n = 60 | ATAAD type-A dissection n = 17 | P-value |
|---|---|---|---|---|
| Male sex | 11/12 (91%) | 41/60 (68%) | 12/17 (70%) | 0.257 |
| Age (years) | 62 [57–71] | 65 [55–70] | 62 [52–69] | 0.732 |
| BSA (m$^2$) | 2.0 [1.9–2.1] | 2.0 [1.8–2.1] | 2.0 [1.9–2.0] | 0.231 |
| Weight (kg) | 85 [81–97] | 80 [72–94] | 80 [75–88] | 0.377 |
| Height (m) | 1.8 [1.7–1.8] | 1.7 [1.7–1.8] | 1.7 [1.7–1.8] | 0.196 |
| Hypertension | 8/12 (66%) | 45/60 (75%) | 16/17 (94%) | 0.156 |
| Diabetes mellitus type 2 | 0/12 (0%) | 6/60 (10%) | 1/15 (6%) | 0.497 |
| Hypercholesterolemia | 7/12 (58%) | 33/60 (55%) | 12/16 (75%) | 0.351 |
| COPD | 0/12 (0%) | 5/60 (8%) | 4/17 (23%) | 0.0852 |
| History of myocardial infarction | 4/12 (33%) | 8/60 (13%) | 3/15 (20%) | 0.235 |
| Family history of CVD | 8/12 (66%) | 21/53 (39%) | 6/10 (60%) | 0.157 |
| Family history of aortic disease[a] | 0/12 (0%) | 12/53 (22%) | 4/9 (44%) | 0.0471 |
| Known genetic mutation[b] | 0/12 (0%) | 6/60 (10%) | 1/17 (5%) | 0.474 |
| Current smokers | 2/12 (16%) | 7/60 (11%) | 5/10 (50%) | 0.0225 |
| Alcohol use | 8/12 (66%) | 40/60 (66%) | 7/7 (100%) | 0.187 |
| Aortic diameter (mm) | 37 [36–40] | 53 [48–59] | 55 [50–60][c] | <0.001 |
| Aortic valve insufficiency | 6/12 (50%) | 51/60 (85%) | 13/16 (81%) | 0.0228 |
| Aortic valve stenosis | 9/12 (75%) | 13/60 (21%) | 1/16 (6.3%) | <0.001 |
| Bicuspid aortic valve | 5/12 (41%) | 20/60 (33%) | 2/16 (12%) | 0.186 |

*Note*: All continuous variables are presented as medians with interquartile ranges [25th percentile to 75th percentile]. All categorical variables are defined as number of subjects and percentage of available data per group. Some of the variables in ATAAD were not fully reported. The independent-samples Kruskal–Wallis test is used to compare continuous variables, whereas the chi-square test is used for categorical variables.

Abbreviations: ATAA, ascending thoracic aortic aneurysm; ATAAD, ATAA dissection; BSA, body surface area; COPD, chronic obstructive pulmonary disease; CVD, cardiovascular disease.

[a] Family history of aortic disease includes aneurysm, dissection, aortic coarctation, aortic valvular disease and connective tissue disease.

[b] Known genetic mutations include *LDS*, *MYH7*, *MYH11*, *EMILIN1*, *SLCA2A* and *TGFBR1*.

[c] Diameters for ATAAD were available in $n = 8$ cases.

previous aortic repair, endocarditis, patients undergoing off-pump surgery and those aged under 18 were not included in the study. Informed consent was obtained from all patients. Demographic information, such as aortic size, sex, body surface area (BSA), hypertension, diabetes, hypercholesterolemia, valve function and valve morphology, was collected (Table 1).

## Tissue collection

In ATAA cases residual aneurysm tissue samples were collected to study tissue characteristics of the maximally dilated region. Although 60 patients were enrolled (Table 1), only 57 were included in the final analysis due to the inadequate tissue quality in the three samples. To secure orientation during further processing the surgeon placed a suture at the anterior cranial edge of the excised ring, as described previously (Ganizada et al., 2023). For the intergroup comparison between ATAA, ATAAD and control groups anterior aortic segments were obtained

(Fig. 1). However if the anterior region was included in the dissection (i.e. in ATAAD cases), tissue samples were taken from the non-dissected region opposing the dissection. Tissue from dissected regions in our ATAAD is not included in our present analysis. In controls either the anterior punch for the proximal anastomosis in CABG patients or a biopsy at the site of aortotomy in AVR patients was collected.

Within the ATAA group the resected aortic ring was sectioned to allow an intragroup comparison between four anatomical regions along the circumference: ventral (AV), medial (AM), lateral (AL) and dorsal (AD) (Fig. 1).

All samples were processed by the investigators within the operating room following established standard operating procedures (SOP) (Ganizada et al., 2023). Every sample was labelled and assigned a code devoid of any patient identifiers. The resected tissues were further analysed for different histomorphometry parameters, such as medial wall thickness, collagen/elastin content, elastin structural integrity, as well as VSMC density, $\alpha$-SMA, calponin, MMP-2 and MMP-9.

## Histology: sample preparation, staining and quantification

For histological examination all tissues were fixed in 1% Hepes buffered formalin solution, adjusted to pH 7.4 at room temperature for 24 h and embedded in paraffin. Cross-sections of aortic tissues (4 μm thick) were stained with haematoxylin and eosin (H&E) to evaluate tissue morphology. Verhoeff–Van Gieson staining was performed to assess elastin content, and sirius red staining was used to identify the fibrillary collagen network of types I and III. To assess the degree of collagen deposition and elastin degradation in the media we quantified the relative percentage of fibre content (expressed as positive staining area per observed area) using image analysis. The VSMC density was assessed from the H&E-stained sections and calculated from the number of detected cell nuclei per observed area (full section).

Immunohistochemical stainings were performed following the laboratory's SOP. After hydration and blocking of endogenous peroxidase activity, sections were incubated with monoclonal antibodies against contractile VSMC markers calponin (CNN1) (1:200; Abcam,

Cambridge, UK) and α-smooth muscle actin (α-SMA) (1:400; Dako; California, CA, USA), human MMP-2 catalytic domain (1:200; Biorbyt, Cambridge, UK), MMP-9 catalytic domain (1:200; Biorbyt), CD68 (1:600, Agilent/DAKO). HRP-conjugated secondary antibodies were visualized with Nova-RED substrate (Vector labs, Amsterdam, The Netherlands). The content percentage was calculated as the positively stained area per observed area.

## Image capture and quantitative analysis

Two-dimensional analysis quantification was facilitated by an Olympus digital slide scanner TH4-200 BX61VS, using a magnification of 10×. Scanned histological slides were analysed by identifying positive staining areas using QuPath (version 0.3.0) (Bankhead et al., 2017). The process of quantification of positive staining percentage included the following steps: image acquisition, selection of a region of interest (ROI) in sectional intima-media layers (adventitia was excluded), adjustment of threshold parameters using a threshold algorithm, measurement of the threshold area, quantification of the percentage

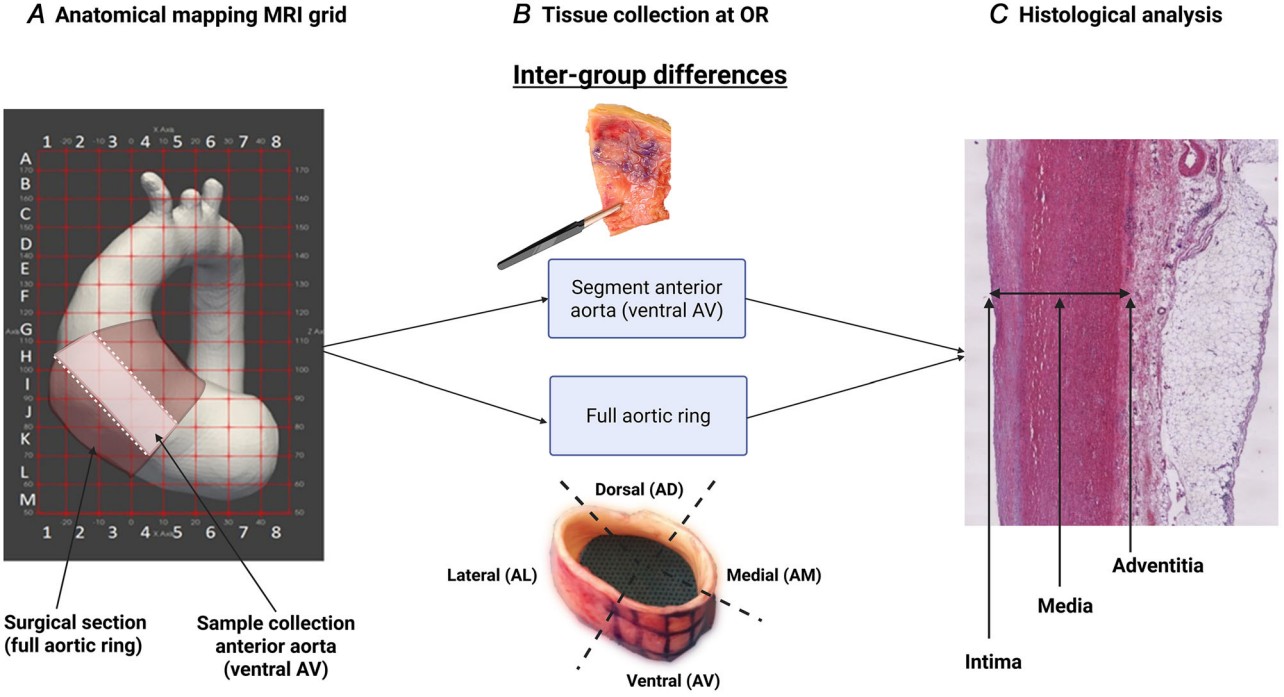

**A** Anatomical mapping MRI grid **B** Tissue collection at OR **C** Histological analysis

**Inter-group differences**

Segment anterior aorta (ventral AV)

Full aortic ring

Dorsal (AD)

Lateral (AL) Medial (AM)

Ventral (AV)

Surgical section (full aortic ring) Sample collection anterior aorta (ventral AV)

Adventitia

Media

Intima

**Intra-patient differences in ATAA**

**Figure 1. Tissue sampling**
*A*, origins of the residual ascending thoracic aortic tissue samples. *B*, anterior samples (ventral AV) were collected for controls, ascending thoracic aortic aneurysm (ATAA) and ATAA dissection (ATAAD) (only the intact anterior, as opposed to the dissected one) to study intergroup histomorphometric differences. In the ATAA cases ventral (AV), medial (AM), lateral (AL) and dorsal (AD) samples were collected to enable the study of intrapatient circumferential histomorphometric differences. *C*, stained tissue section illustrating intima, media and adventitia tissue layers. [Colour figure can be viewed at wileyonlinelibrary.com]

staining in the ROI and imaging the ROI area. The same parameters and thresholds were used for all the images from each staining. Positive pixels were expressed as percentages of the total media tissue area.

### Histomorphometric analysis

Morphometric analyses were conducted using QuPath (version 0.3.0) software on Verhoeff–Van Gieson-stained sections (Bankhead et al., 2017). The tissue's intima-media thickness (IMT) was measured at three specific locations along the longitudinal axis perpendicular to the lumen (excluding the adventitia). The mean of these three IMT measurements was utilized for the analysis.

Elastin laminar thickness and interlaminar distance were also estimated using QuPath and estimated as the mean of 10 measurements per section along the longitudinal axis perpendicular to the lumen.

Elastin laminar breakage was assessed using a custom-built MATLAB (R2024a, MathWorks, Natick, MA, USA) script. Briefly first an ROI was selected considering the medial layer; then a binary image was created by thresholding to distinguish staining from non-stained areas. The binary image was then transformed into a skeleton to identify laminar endpoints and determine end-to-end length. A laminar break was defined as a discontinuity of the linear intact elastin fibres (Verbrugghe et al., 2018). For each sample elastin laminar breakage was quantified by the formula:

$$\text{Elastin laminar breakage} = \frac{n \text{ of detected endpoints}}{\text{total elastin segment length in } \mu\text{m}},$$

in $[\mu\text{m}^{-1}]$.

### Statistical analysis

Statistical analysis was performed using GraphPad Prism 10 (GraphPad Software, San Diego, CA, USA).

For continuous variables normality was assessed using the Shapiro–Wilk test, indicated by $P$-values $> 0.05$. All normally distributed data are presented as mean $\pm$ SD along with corresponding $n$ values. Skewed data (Shapiro–Wilk $P$-value $\leq 0.05$) are presented as a median with interquartile ranges [25th percentile to 75th percentile].

Intergroup comparisons for skewed data were performed using Kruskal–Wallis testing followed by Dunn's test for pairwise comparisons. Intergroup comparisons for normal data were performed using one-way ANOVA followed by Tukey's multiple comparisons tests. For intergroup comparisons (control, ATAA, ATAAD), each data point in the swarm and correlation plots represents an independent biological sample from a single patient (i.e. $n$ equals the number of patients per group).

For intragroup comparisons among the four anatomical regions within the ATAA group (see Fig. 1) repeated-measures ANOVA with Geisser-Greenhouse correction was used for equal variability of differences. *Post hoc* pairwise comparisons were conducted using Tukey's multiple comparisons test.

For intragroup comparisons for each patient four circumferential measurements (AV, AM, AL, AD) were taken from the excised tissue.

Chi-squared tests were used to compare categorical variables between groups, whereas Spearman's correlation was applied to assess relationships between quantitative histological data. All statistical tests were two-tailed, and a $P$-value of $<0.05$ was considered statistically significant.

## Results

### Baseline characteristics

The median diameter of the ascending aorta was 53 mm in ATAA patients, 55 mm in ATAAD and 37 mm in the control group (Table 1). In all three groups male was the predominant sex. Among the subjects with ATAAD there was a higher percentage of chronic obstructive pulmonary disease (COPD), smoking and alcohol consumption than in control and ATAA patients. Among ATAA cases 6 patients had a known genetic mutation (*LDS*, *MYH7*, *MYH11*, *EMILIN1* and *SLCA2A*), whereas 1 case of ATAAD was associated with a *TGFBR1* mutation. There were no significant differences between groups in other determinants that may influence aortic size, such as sex, weight, age, BSA, hypertension, hypercholesterolemia and valve morphology (Table 1).

### Histomorphometric differences between non-dilated, aneurysmal and dissected aortas

**Intima-media thickness and aortic diameter.** Pre-operative maximum ascending aortic diameter (Fig. 2*A*,*C*) was different between ATAA and control groups ($P < 0.0001$; $n = 58$ and $n = 12$, respectively), ATAAD and control groups ($P < 0.0001$; $n = 8$ and $n = 12$, respectively) but not between ATAA and ATAAD groups ($P = 0.935$; $n = 58$ and $n = 8$, respectively). We observed a significant difference in the IMT of the anterior aorta (Fig. 2*B*,*D*) between patients with ATAA and controls ($P = 0.00320$; $n = 57$ and $n = 12$, respectively), as well as between ATAA and ATAAD ($P = 0.0189$; $n = 57$ and $n = 17$, respectively). No significant difference in IMT was observed between ATAAD and control ($P = 0.999$; $n = 17$ and $n = 12$, respectively). It should be mentioned that the measurement of IMT excluded the adventitial layer for all three groups (Fig. 2*B*). Substantial intragroup

variability in IMT was evident within the ATAA cohort, as reflected by a coefficient of variation of 25.5%. The relation between aortic diameter and IMT across three groups (control, ATAA and ATAAD) is further examined in Fig. 3.

**Ratio between aortic radius and thickness.** The distribution of the IMT within the three groups of patients shows a wide range in the ATAA group compared to the ATAAD and control groups, as shown in Fig. 2. To assess the extent of geometric adaptation to wall stress in ATAA the ratio between aortic radius (diameter/2) and IMT was calculated (Fig. 3). The scatterplot in Fig. 3 also includes, as a reference, the average IMT values for the ATAAD and control groups. The results reveal a notable variability in the radius-to-IMT ratio, suggesting

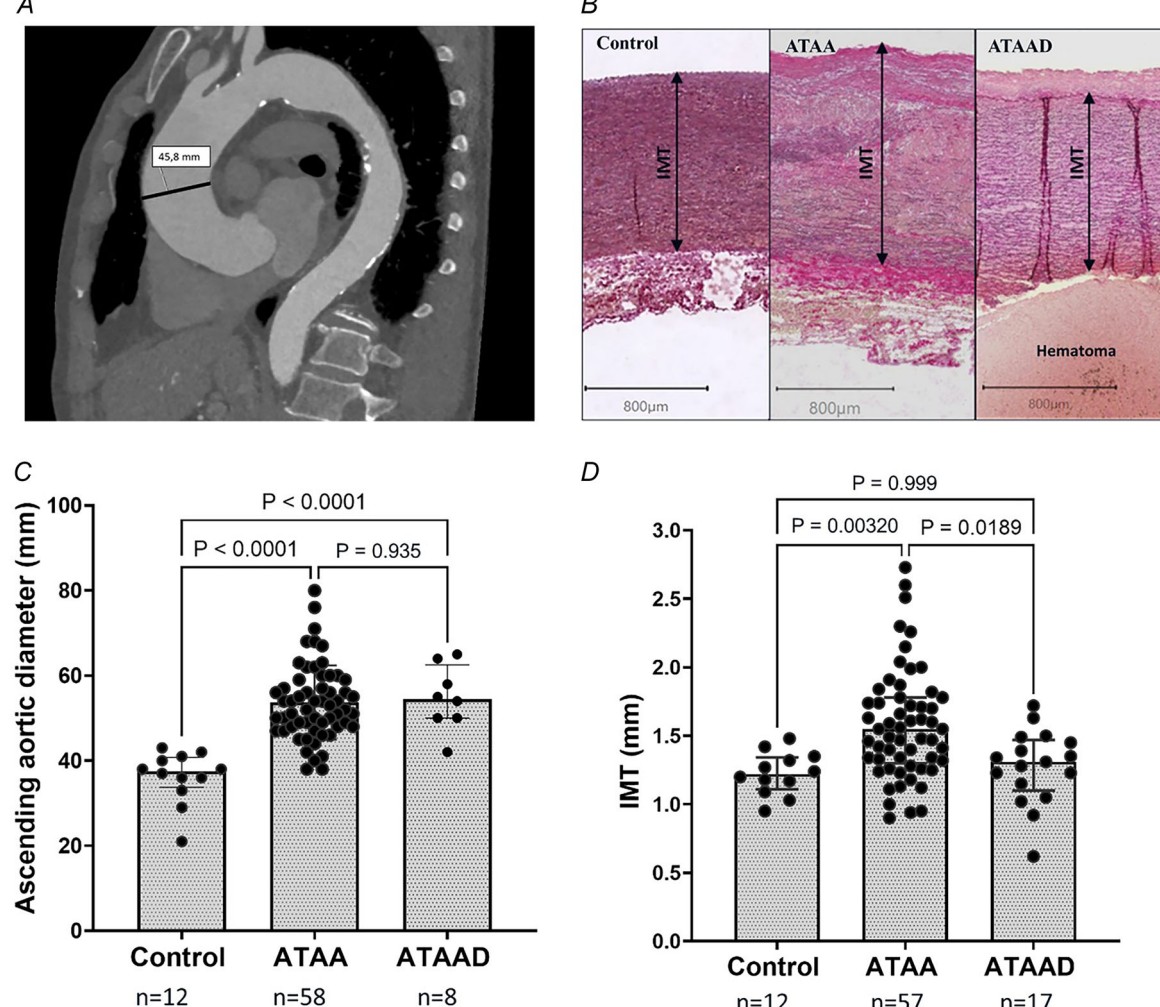

**Figure 2.** *A, preoperative ascending aortic diameter was measured using computed tomography (CT) imaging (single measurement as indicated in panel (A). B, histological assessment of intima-media thickness (IMT) of the anterior aortic wall was performed at 10× magnification*
Haematomas were excluded from IMT measurements in ascending thoracic aortic aneurysm dissection (ATAAD) cases. The subject-level IMT value reflects the mean of three measurements per sample; see Section 2.5 (in panel (*B*) only one of the three is illustrated). Due to limited preoperative CT availability aortic diameter data were obtained from 8 of 17 ATAAD patients. IMT data were available for 57 of 60 patients (3 excluded due to poor tissue quality; Section 2.2). Statistical analysis was conducted using one-way ANOVA with Tukey's *post hoc* test for normally distributed ascending aortic diameter; Kruskal–Wallis test for non-normally distributed IMT. *C*, ascending aortic diameter was significantly increased in ATAA and ATAAD compared to controls ($P < 0.0001$), with no difference between ATAA and ATAAD ($P = 0.935$). *D*, IMT was significantly greater in ATAA compared to controls ($P = 0.0032$) and ATAAD ($P = 0.0189$); no difference between ATAAD and controls ($P = 0.999$). Mean ± SD of ascending aortic diameter: control: $36.17 \pm 6.16$ mm; ATAA: $53.66 \pm 8.74$ mm; ATAAD: $54.75 \pm 7.65$ mm. Median [25th percentile to 75th percentile] for IMT: control 1.22 [1.11–1.34] mm; ATAA 1.55 [1.30–1.78] mm; ATAAD 1.31 [1.10–1.47] mm. [Colour figure can be viewed at wileyonlinelibrary.com]

roughly two distinct patterns of aortic wall thickness adaptation in response to dilatation, with half of the ATAA group tending towards 'thickening' (indicative of partially intact wall stress homeostasis) and the other half tending towards 'thinning' (indicative of loss of wall stress homeostasis).

**Extracellular matrix remodelling.** The current findings indicate a significant decrease in the medial fraction of collagen (%) (Fig. 4*B*) in patients with ATAA and ATAAD ($P = 0.0204$; $n = 57$ and $P < 0.001$; $n = 17$, respectively) compared to controls ($n = 12$), with a significant decline observed in ATAAD compared to ATAA ($P = 0.0383$; $n = 17$ and $n = 57$, respectively). In addition to collagen the fraction of elastin (%) (Fig. 4*C*) was significantly lower in ATAA ($P < 0.00001$; $n = 57$) and ATAAD ($P = 0.00897$; $n = 17$) compared to controls ($n = 12$). Both elastin and collagen contents are reduced in ATAA(D) compared to controls, as illustrated by representative histological images (Fig. 4*A*).

**Compromised elastic lamellar network.** As reported above quantitative analysis revealed a significant decrease in the fraction of elastin (%) in ATAA(D) patients compared to controls. To assess the degree of elastin fragmentation lamella thickness and interlamellar distance of elastin fibres were calculated within all groups

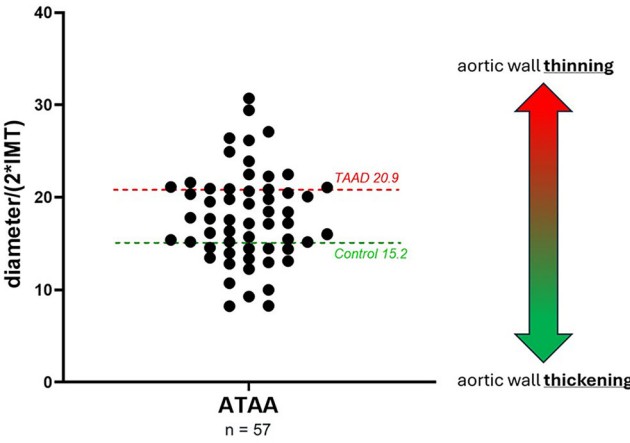

**Figure 3. Significant variability in aortic wall adaptation among patients with ascending thoracic aortic aneurysm (ATAA)**
The diagram illustrates the aortic radius-to-thickness ratio, calculated from the measured aortic diameter and the anterior intima-media thickness (IMT) of ATAA dissection (ATAAD) and control patients. In ATAA patients a ratio >20.85 (calculated from group averages of diameter and IMT in ATAAD) signifies relative wall thinning akin to dissection cases. A ratio <15.16 (calculated from group averages of diameter and IMT in controls) indicates relative thickening, suggesting intact wall stress homeostasis. Each data point represents an individual patient, with the mean ATAA value reported as 17.92 ± 4.99 (mean ± SD). [Colour figure can be viewed at wileyonlinelibrary.com]

(Fig. 5*A*–*E*). The results reveal that lamella thickness in ATAA and ATAAD was significantly lower ($P < 0.0001$; $n = 57$ and $P < 0.0001$; $n = 17$, respectively) compared to controls ($n = 12$) (Fig. 5*B*). Although no difference in alteration in the fraction of elastin was found between ATAA and ATAAD, there was a significant decrease in lamella thickness observed in ATAAD patients compared to ATAA ($P = 0.00420$; $n = 17$ and $n = 57$, respectively) (Fig. 5*B*). Further the interlamellar distance was significantly increased in both ATAA and ATAAD ($P < 0.0001$; $n = 57$ and $n = 17$, respectively) compared to the controls ($n = 12$) (Fig. 5*C*). Concordant with lamella thickness interlamellar distance in ATAAD patients was significantly higher compared to ATAA ($P = 0.00510$; $n = 17$ and $n = 57$, respectively) (Fig. 5*C*). This indicates that ECM is severely degenerated in ATAAD patients, especially the elastic lamellar unit. Additionally there is a considerable overlap in abnormalities between the ATAA and ATAAD groups, considering the ECM degeneration grade. In addition to variations in lamellar thickness and spacing significant lamellar disruptions or breaks were observed in both ATAA and ATAAD groups ($P < 0.0001$; $n = 44$ and $n = 10$, respectively) compared to controls ($n = 10$) (Fig. 5*D*). No significant difference was observed between the ATAA and ATAAD groups ($P = 0.462$; $n = 44$ and $n = 10$, respectively). A representative histological image is shown in Fig. 5*E*. These findings not only indicate a reduction in elastin content in ATAAD but also emphasize significant structural changes in the aortic wall, suggesting an association between elastin degradation and tissue architecture alterations. This is represented in the Spearman correlation graphs in Fig. 6*A,B*.

**Morphology and increased density of VSMCs.** The number of VSMC nuclei was detected for the selected regions of interest from H&E-stained sections. The cell nuclei detection was assessed by calculating the average number of cell nuclei per unit tissue surface area (cell nuclei/mm$^2$). Despite overall medial degeneration, as shown in previous sections (3.2.3 and 3.2.4), no loss of VSMCs was observed in the medial layer of aneurysmatic aortas (Fig. 7*A*). In patients with ATAA and ATAAD the aortic media wall showed an increased number of VSMCs per cross-sectional area compared to controls, with significant differences in ATAA ($P = 0.0220$; $n = 57$, controls $n = 10$) and ATAAD ($P < 0.001$; $n = 17$, controls $n = 10$) cases. Additionally a slight increase in VSMCs was noted in ATAAD compared to ATAA ($P = 0.0343$; $n = 17$ and $n = 57$, respectively) (Fig. 7*B*). The VSMCs increased in number and displayed varied morphological shapes. In controls VSMCs exhibit spindle-like morphology, whereas in aneurysmal or dissected aortas VSMCs transmute into a rhomboid shape (Fig. 7*A*). Furthermore there was a positive correlation ($\rho = 0.351$) between

VSMC density and interlamellar distance (Fig. 8). This correlation suggests a potential compensatory mechanism involving a shift towards the synthetic VSMC phenotype in response to elastin degradation.

**Decreased expression of VSMC contractile markers.** VSMC contractile proteins, calponin and $\alpha$-SMA, were detected by immunohistochemical staining. The positive pixel percentage of calponin and $\alpha$-SMA was calculated as a fraction of the total surface area of the anterior aorta considered in the analysis (Fig. 9). Alongside the change in VSMC morphology and migratory cell response

patients with ATAA and ATAAD exhibited a local increase in VSMC number, which was accompanied by a local reduction in contractile markers in the medial layer of the aortic wall (Fig. 9A). Calponin was significantly reduced in ATAA ($P < 0.001$; $n = 52$) and ATAAD ($P < 0.0001$; $n = 11$) compared to controls ($n = 11$) and slightly reduced in ATAAD compared to ATAA ($P = 0.0170$; $n = 11$ and $n = 52$, respectively) (Fig. 9B). The $\alpha$-SMA marker also showed a significant decrease in ATAA ($P = 0.00330$; $n = 53$) and ATAAD ($P < 0.001$; $n = 13$) compared to controls ($n = 11$), with no significant difference between ATAAD and ATAA ($P = 0.213$; $n = 53$ and $n = 13$, respectively) (Fig. 9C).

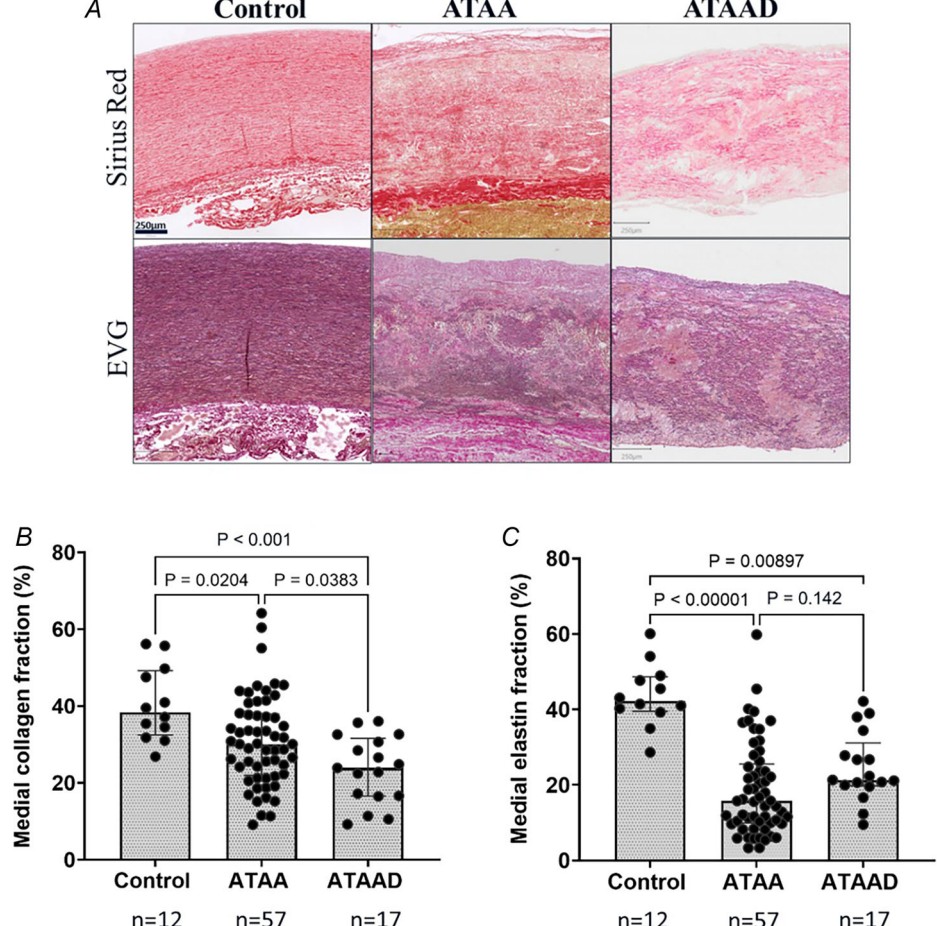

**Figure 4. Compared to controls, both elastin and collagen contents decrease in the anterior aorta of ascending thoracic aortic aneurysm (dissection) (ATAA)(D)**
*A*, transverse sections of the ascending aorta from control, ATAA and ATAAD patients. The sections are stained for collagen (sirius red) and elastin (elastic Van Gieson (EVG)) and imaged at 10× magnification. One-way ANOVA with Tukey's *post hoc* test was used for normally distributed collagen fraction; Kruskal–Wallis test was applied to non-normally distributed elastin fraction. *B*, medial fraction of collagen (%) showed a significant difference between ATAA–control ($P = 0.0204$), ATAAD–control ($P < 0.001$) and ATAA–ATAAD ($P = 0.0383$). *C*, the fraction of elastin (%) showed a significant difference between ATAA–control ($P < 0.00001$) and ATAAD–control ($P = 0.00897$) and no difference between ATAA and ATAAD ($P = 0.142$). The mean ± SD for collagen: control: 40.54 ± 9.71; ATAA: 31.00 ± 11.76; and ATAAD: 23.43 ± 8.72. The median [25th percentile to 75th percentile] for elastin: control: 42.29 [39.52–48.66]; ATAA: 15.79 [9.91–25.55]; and ATAAD: 21.32 [19.79–31.14]. [Colour figure can be viewed at wileyonlinelibrary.com]

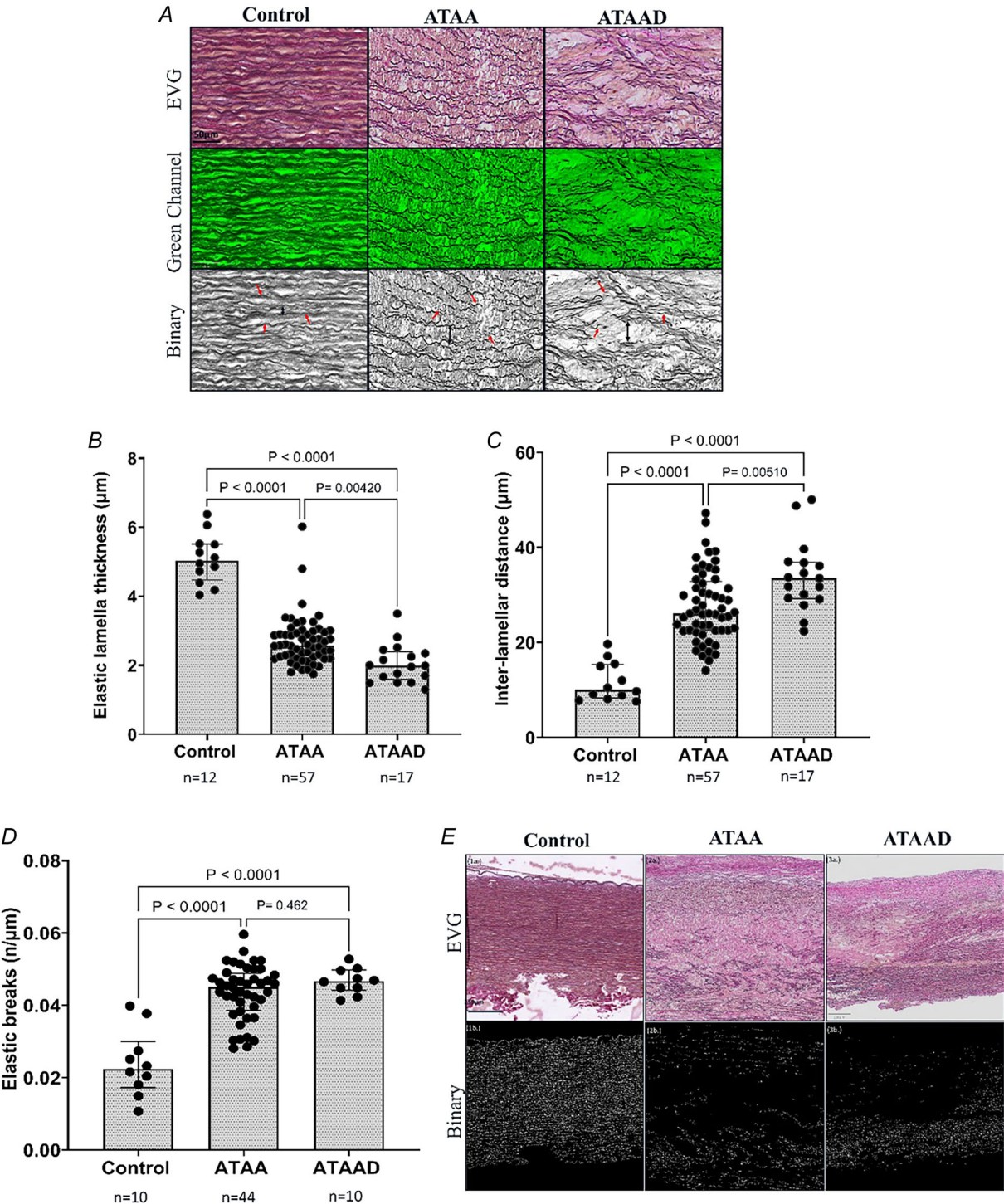

**Figure 5. Pronounced deterioration of elastin lamella was observed in the ascending thoracic aortic aneurysm (ATAA) group, which was further exacerbated in ATAA dissection (ATAAD)**

*A*, transverse sections of the anterior aorta from control, ATAA and ATAAD groups were stained with elastic Van Gieson (EVG) stain for elastic lamella at a magnification of 10×. For enhanced visualization of elastic fibres a green channel was used, and structural disruptions binarized channel. Note: Due to validation of MATLAB set-up for elastic lamella breaks a limited number of samples were analysed: control (10/12), ATAA (44/57) and ATAAD (10/17). For elastic lamella thickness, which was not normally distributed, the Kruskal–Wallis test was applied. One-way ANOVA with Tukey's *post hoc* test was used for normally distributed interlamellar distance and elastic

breaks. Elastic lamella thickness showed significant differences: ATAA–control ($P < 0.0001$), ATAAD–control ($P < 0.0001$), ATAA–ATAAD ($P = 0.00420$). *B*, interlamellar distance displayed significant variations: ATAA–control ($P < 0.001$), ATAAD–control ($P < 0.0001$) and ATAA–ATAAD ($P = 0.00510$). *D/E*, elastic lamella breaks, calculated using MATLAB and corrected for lamella length, showed significant differences between ATAA–control ($P < 0.0001$) and ATAAD–control ($P < 0.0001$) but no difference between ATAAD and ATAA ($P = 0.462$). The median [25th percentile to 75th percentile] for control: 5.04 [4.48–5.52]; ATAA: 2.58 [2.23–2.98]; ATAAD: 1.99 [1.59–2.40]. The mean $\pm$ SD for interlamellar distance: control: 11.75 $\pm$ 4.08; ATAA: 27.65 $\pm$ 7.39; and ATAAD: 33.93 $\pm$ 7.38, and for an elastic break: control 0.024 $\pm$ 0.0092; ATAA 0.044 $\pm$ 0.0075; ATAAD 0.047 $\pm$ 0.0035. [Colour figure can be viewed at wileyonlinelibrary.com]

Because of the high-density regions of nuclei in the media, we briefly explored the possible infiltration of phagocytes. However CD68 staining was negative, confirming the absence of macrophages in these high-density areas (data not shown).

Figure 10 presents a subset of Spearman's correlations between cell-matrix data. Contractile markers correlated negatively with VSMC density while showing a positive correlation with elastin content. These findings suggest that VSMCs may migrate to sites of elastin degradation accompanied by a phenotypic shift.

Additionally in the ATAA group the radius-to-IMT ratio did not exhibit a significant correlation with VSMC density or alterations in calponin or $\alpha$-SMA expression in the media (Fig. 11).

**Proteolytic profile of matrix metalloproteinases.** The relative protein expression levels of both MMP-2 and MMP-9 (Fig. 12*A*) significantly increased in tissues from patients with ascending thoracic aortic aneurysm (ATAA) (MMP-2: $P = 0.0151$; MMP-9: $P = 0.0444$, $n = 10$, respectively) and ATAA dissection (ATAAD) (MMP-2: $P = 0.00860$; MMP-9: $P = 0.00160$; $n = 9$, respectively)

compared to controls ($n = 9$) (Fig. 12*B*,*C*). However no significant differences were observed between the ATAA and ATAAD groups for either MMP-2 ($P > 0.999$, $n = 10$ and $n = 9$, respectively) or MMP-9 ($P = 0.788$; $n = 10$ and $n = 9$, respectively) (Fig. 12*B*,*C*). These study findings suggest that the dysregulation of MMP-2 and MMP-9, two key enzymes involved in the degradation of the ECM within the aortic wall, may play a crucial role in the pathogenesis of thoracic aortic aneurysms and dissections. All findings are further summarized in Table 2 for reference.

### No histological differences along the aneurysm circumference

Multiple paired-ANOVA testing within the ATAA group showed no statistical differences in IMT, collagen content, elastin content, VSCM density or phenotype markers along the aneurysm circumference ($n = 52$). For a corresponding illustrative case of histological samples along the circumference please refer to Fig. 13.

Upon this striking result we performed the same analysis for 'thickening' and 'thinning' ATAA sub-

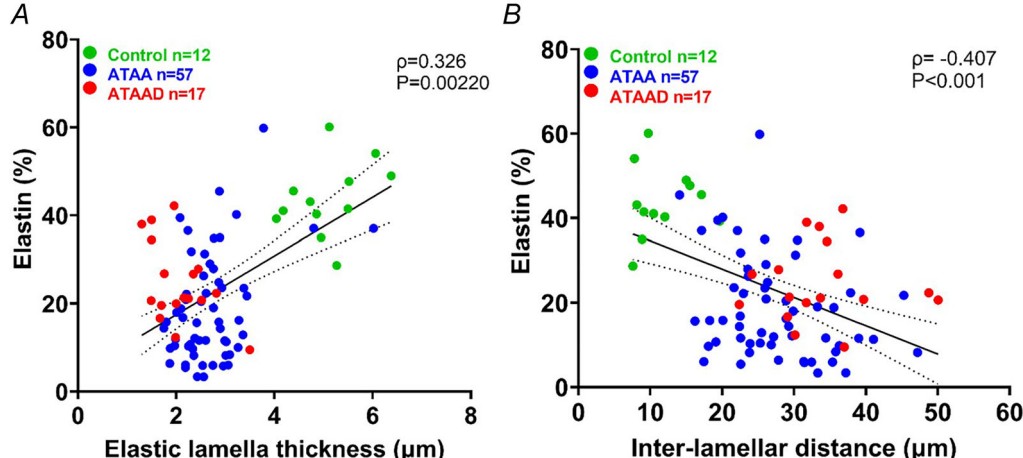

**Figure 6. Association between elastin quantity and lamella quality in the anterior aorta**
Control (green), ATAA (blue) and ATAAD (red) were analysed by histopathology and histomorphometry. The media fraction of elastin, lamella thickness and interlamellar distance were quantified from the elastic Van Gieson (EVG) stain. *A*, Spearman's correlation revealed a positive association between elastin percentage and elastic lamella thickness ($\rho = 0.326$), with a significant difference between the groups ($P = 0.00220$). *B*, a negative association between elastin% and interlamellar distance ($\rho = -0.407$) and significant difference between the groups ($P < 0.001$). [Colour figure can be viewed at wileyonlinelibrary.com]

groups by dichotomizing around the median value of the radius-to-IMT ratio (ref. Fig. 3). The results are given in Fig. 14. In the thickening subgroup ($n = 27$, Fig. 14*A*) there were no statistical differences along the circumference. In the thinning subgroup we also found no conspicuous differences at the cell-matrix level but did

find ventral IMT to be statistically lower than the other three segments ($n = 25$, Fig. 14*B*).

Considering the potential for selection bias because thickening and thinning groups were defined using IMT measurements from ventral (AV) samples (ref. Fig. 3), we also conducted a stratified analysis based on a dichotomy

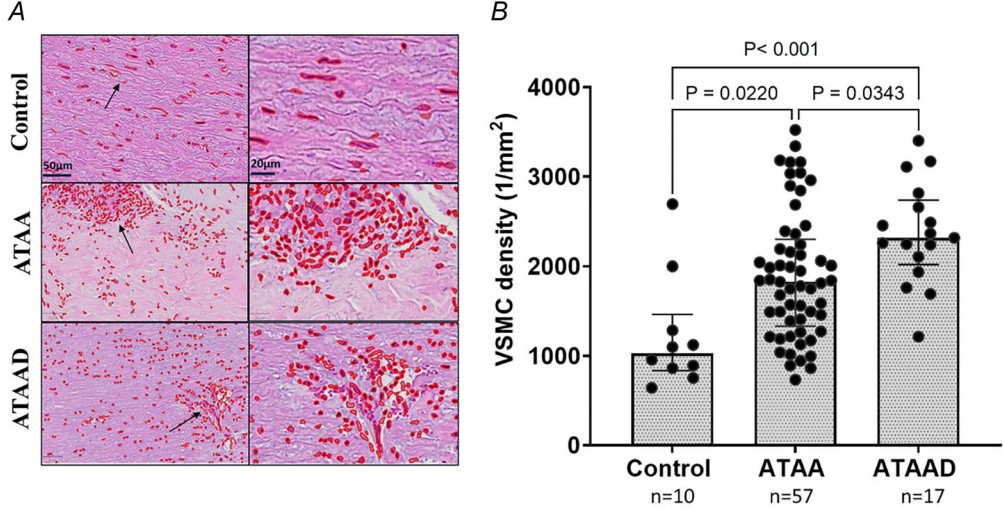

**Figure 7. Increased vascular smooth muscle cell (VSMC) density observed in ascending thoracic aortic aneurysm (dissection) (ATAA)(D) patients**
*A*, histological images of the anterior aortas from control, ATAA and ATAAD patients, stained with haematoxylin and eosin to detect VSMC cells (digital cell nuclei detection marked in red) at 10× magnification. Statistical analysis was performed using the Kruskal–Wallis test. *B*, the total number of VSMCs per cross-sectional area (cell nuclei/mm$^2$) showed significant differences between ATAA–control ($P = 0.0220$), ATAAD–control ($P < 0.001$) and ATAA–ATAAD ($P = 0.0343$). Median [25th percentile to 75th percentile] values for VSMC density were as follows: control 1027 [835–1463]; ATAA 1827 [1331–2300]; and ATAAD 2316 [2020–2736]. [Colour figure can be viewed at wileyonlinelibrary.com]

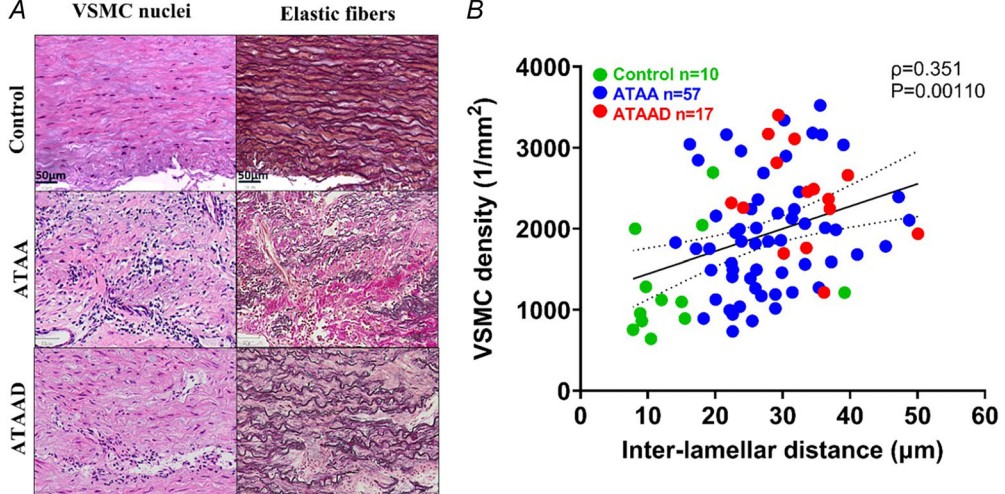

**Figure 8. Association between elastin loss and elevated vascular smooth muscle cell (VSMC) density**
*A*, elastin degradation is compensated by the migration of VSMCs to the area of elastin loss, as evidenced by the formation of VSMC cell clusters in the anterior aorta. *B*, control aortas from non-dilated patients (green), ascending thoracic aortic aneurysm (ATAA) (blue) and ATAA dissection (ATAAD) (red). Spearman's correlation revealed a positive association between elastin percentage and VSMC density ($\rho = 0.351$) with a significant difference between the groups ($P = 0.00110$). [Colour figure can be viewed at wileyonlinelibrary.com]

using the median of the radius-to-IMT ratio based on the average of segmental IMTs along the circumference. A significant difference between ventral and dorsal IMT remained ($P = 0.0190$, not shown), but other differences no longer achieved statistical significance, confirming some selection bias in the former analysis but also substantiating more heterogeneity in IMT in thinning aneurysms.

## Discussion

The pathological remodelling of the ascending aorta in aneurysm patients is still debated, as ATAAD is not always preceded by aneurysm formation. To assess the range of aortic wall remodelling in aneurysm patients we compared local histopathological changes between patients with aortic aneurysms, dissections and control patients without aortic disease. We found the following key observations between groups (Table 2): (1) medial thickening coincides with dilatation in roughly half of the aneurysm repair cases but not in dissection and remaining repair cases; (2) collagen and elastin content are decreased in aneurysms and dissection patients, clearly accompanied by a loss of elastin structural integrity associated with increased proteolysis; (3) medial collagen content is significantly decreased in type-A dissected

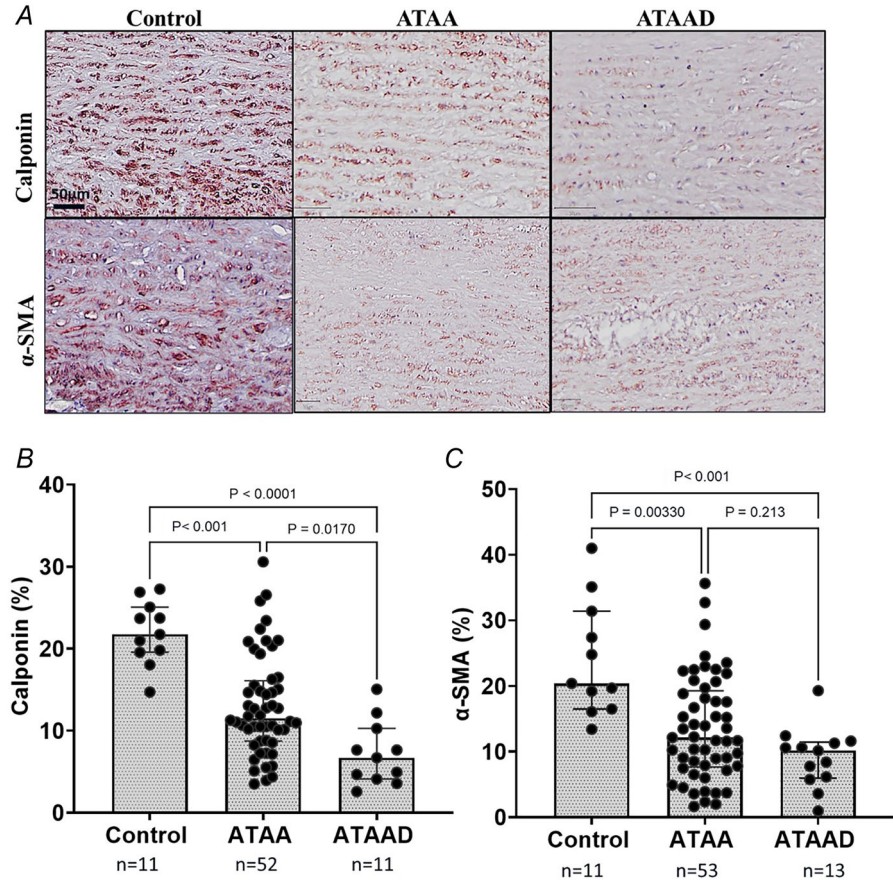

**Figure 9. A marked reduction in contractile vascular smooth muscle cell (VSMC) markers was observed in ascending thoracic aortic aneurysm (dissection) ATAA(D)**
*A*, histological representation of anterior aortic tissues obtained from control, ATAA and ATAAD aortas stained with calponin and $\alpha$-smooth muscle actin ($\alpha$-SMA) antibody stain, with magnification 10×. Note: 5/57 ATAA patient samples were excluded from the calponin analysis, and 4/57 ATAA patient samples were excluded from the $\alpha$-SMA analysis due to poor tissue quality. Statistical tests were performed using Kruskal–Wallis test. *B*, relative protein expression of calponin was significantly lower in ATAA–control ($P < 0.001$), ATAAD–control ($P < 0.0001$) and ATAAD–ATAA ($P = 0.0170$). *C*, relative protein expression of $\alpha$-SMA was significantly lower in ATAA–control ($P < 0.01$) and ATAAD–control ($P < 0.001$), with no difference between ATAAD and ATAA ($P = 0.213$). Median [25th percentile to 75th percentile] values for calponin: control: 21.76 [19.56–25.07]; ATAA: 11.52 [8.75–16.10]; and ATAAD: 6.71 [4.13–10.28]; and for $\alpha$-SMA: control: 20.40 [16.50–31.40]; ATAA: 12.14 [7.64–19.27]; and ATAAD: 10.16 [5.97–11.45]. [Colour figure can be viewed at wileyonlinelibrary.com]

cases compared to aneurysm repair cases; (4) matrix degradation is accompanied by increased VSMC density and loss of contractile phenotype; (5) within 'thickening' aneurysms – that is along the circumference – we found no differences in media IMT, collagen and elastin content or density or phenotype of VSMCs; and (6) within 'thinning' aneurysms we found little heterogeneity; however we observed that IMT was more variable along the circumference.

## Wall stress homeostasis may be (partially) intact in aneurysm formation

Previous studies have reported conflicting results when comparing aortic wall thickness between ATAA and ATAAD (Iliopoulos et al., 2009; Shiran et al., 2014; Van Puyvelde et al., 2016). We found that the ATAA patients show a wide range of medial wall thickness, whereas in the ATAAD group medial wall thickness is significantly smaller and rather comparable to control aortas. It is tempting to speculate that the thinning of the medial wall is a consequence of the increase in absolute aortic diameter (assumed constancy of wall volume), but this is not the case for all ATAA cases. The increased IMT observed in a substantial fraction of ATAA patients could well indicate adaptation to normalize mechanical wall stress (Humphrey & Schwartz, 2021). Our findings are corroborated by Zhang et al., showing that the increase in IMT serves as an adaptive response for aortic mechanical homeostasis in mice, with YAP signalling as a key driving factor (Zhang et al., 2023).

A previous study, in which we found wall stress normalization and homogenization to be essential in capturing age-related arterial stiffening (Reesink & Spronck, 2019), corroborates the lack of circumferential differences in our ATAA cases: even during the

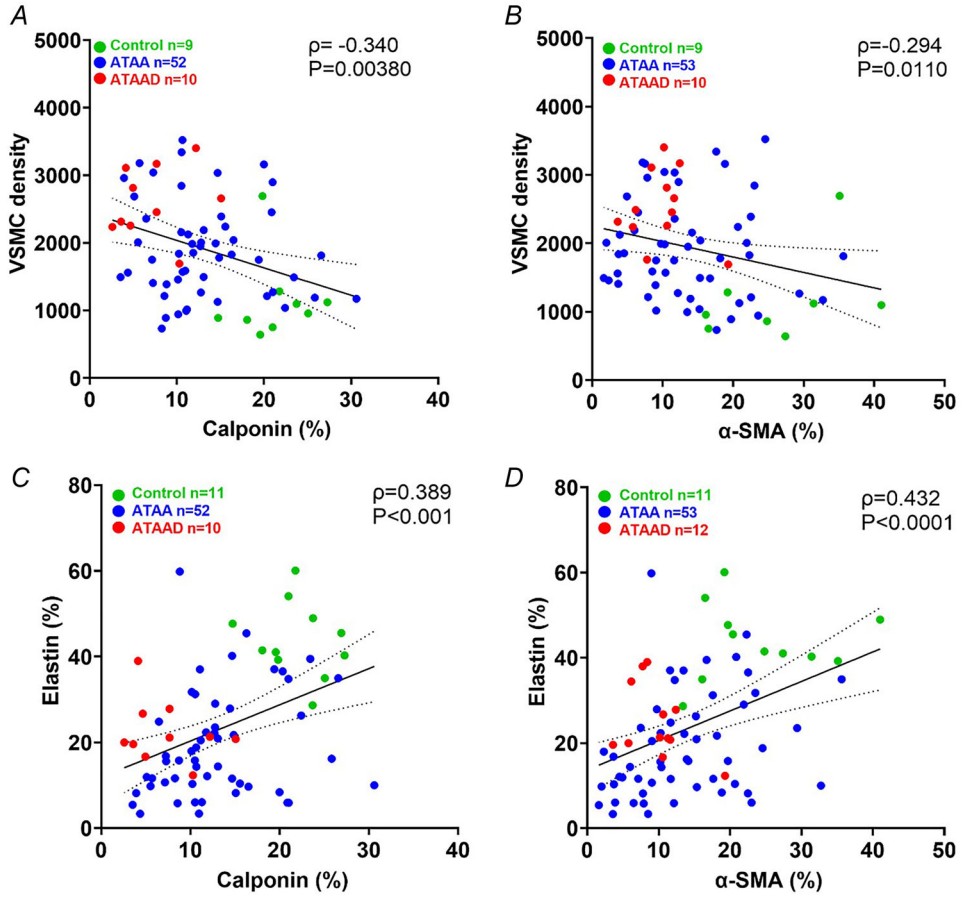

**Figure 10. VSMC density and elastin content exhibit opposing correlations in vascular remodelling**
Spearman's correlation (*A*, *B*) demonstrates a negative correlation with an increase in vascular smooth muscle cell (VSMC) density, whereas parts (*C*) and (*D*) exhibit a positive correlation with an increase in elastin content. Individual data points represent controls (*green*), ascending thoracic aortic aneurysm (ATAA) (*blue*) and ATAA dissection (ATAAD) (*red*). Spearman's correlation analysis revealed a significant negative association between VSMC density and calponin% ($\rho$ = −0.340, $P$ = 0.0038) and $\alpha$-smooth muscle actin ($\alpha$-SMA) ($\rho$ = −0.294, $P$ = 0.0110), whereas elastin% showed a significant positive correlation with both calponin% ($\rho$ = 0.389, $P$ < 0.01) and $\alpha$-SMA ($\rho$ = 0.432, $P$ < 0.0001). [Colour figure can be viewed at wileyonlinelibrary.com]

**Table 2. Summary of key findings on group differences**

| Category variables | | ATAA *vs.* control | ATAAD *vs.* control | ATAA *vs.* ATAAD |
|---|---|---|---|---|
| Morphometric measures | Aortic diameter (mm) | ↑ ($P < 0.0001$) | ↑ ($P < 0.0001$) | ↔ ($P = 0.935$) |
| | IMT(mm) | ↑ ($P = 0.00320$) | ↔ ($P = 0.999$) | ↑ ($P = 0.0189$) |
| ECM content | Collagen% | ↓ ($P = 0.0204$) | ↓ ($P < 0.001$) | ↑ ($P = 0.0383$) |
| | Elastin% | ↓ ($P < 0.00001$) | ↓ ($P = 0.00897$) | ↔ ($P = 0.142$) |
| Elastin quality | Elastin lamella thickness (μm) | ↓ ($P < 0.0001$) | ↓ ($P < 0.0001$) | ↑ ($P = 0.00420$) |
| | Elastin interlamellar distance (μm) | ↑ ($P < 0.0001$) | ↑ ($P < 0.0001$) | ↓ ($P = 0.00510$) |
| | Elastic lamella breaks (n) | ↑ ($P < 0.0001$) | ↑ ($P < 0.0001$) | ↔ ($P = 0.462$) |
| Proteases | MMP-2% | ↑ ($P < 0.05$) | ↑ ($P < 0.01$) | ↔ ($P > 0.05$) |
| | MMP-9% | ↑ ($P < 0.05$) | ↑ ($P < 0.01$) | ↔ ($P > 0.05$) |
| Cell density and contractile phenotype | VSMC density (1/mm$^2$) | ↑ ($P = 0.0220$) | ↑ ($P < 0.001$) | ↓ ($P = 0.0343$) |
| | Calponin% | ↓ ($P < 0.001$) | ↓ ($P < 0.0001$) | ↑ ($P = 0.0170$) |
| | $\alpha$-SMA% | ↓ ($P = 0.00330$) | ↓ ($P < 0.001$) | ↔ ($P = 0.213$) |

Abbreviations: ↑, increased; ↓, decreased; ↔, unchanged; ATAA, ascending thoracic aortic aneurysm; ATAAD, ATAA dissection; IMT, intima-media thickness; MMP, matrix metalloproteinase; $\alpha$-SMA, $\alpha$-smooth muscle actin; VSMC, vascular smooth muscle cell.

pathological dilatation leading to an aneurysm wall thickness and cell-matrix properties remain homogeneous (Fig. 14). Therefore even more striking was our observation that in 'thinning' ATAA – which we interpret as being indicative of greater loss of wall stress homeostasis – cell-matrix properties were circumferentially homogenous despite a clear indication of IMT to be more variable in these thinner aneurysms. Local differences in blood flow have been charted very well in 4D-flow magnetic resonance (MR) studies and could certainly

affect wall stress homeostasis by an overriding demand on VSMCs to relax through the sustained release of endothelial relaxing factors (Ganizada et al., 2023; van Ooij et al., 2021).

Nonetheless considering the statistical variability there exist interindividual differences in medial wall thickness, collagen and elastin levels and VSMC density across the aortic circumference with underlying correlations. Each aneurysm may have its unique path of development, with adaptive and maladaptive responses working in

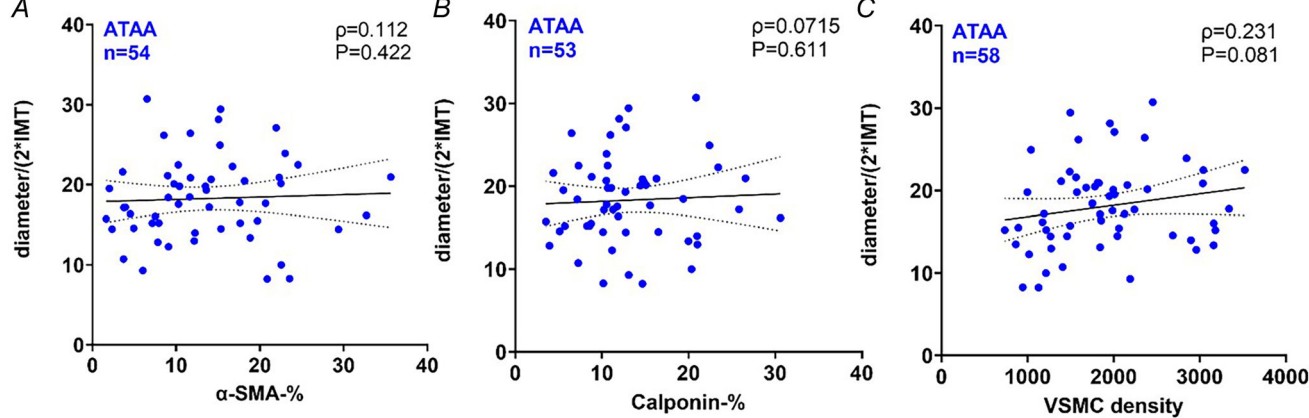

**Figure 11. Cellular markers show no significant association with diameter-to-IMT ratio in ATAA patients**
Spearman's correlation analysis between the diameter-to-intima-media thickness (IMT) ratio and (*A*) $\alpha$-smooth muscle actin ($\alpha$-SMA)% ($\rho = 0.112$, $P = 0.422$), (*B*) calponin% ($\rho = 0.0715$, $P = 0.611$) and (*C*) vascular smooth muscle cell (VSMC) density ($\rho = 0.231$, $P = 0.081$) in patients with ascending thoracic aortic aneurysm (ATAA). Each dot represents an individual ATAA patient. No significant associations were observed in any of the parameters. [Colour figure can be viewed at wileyonlinelibrary.com]

parallel. Correspondingly we deem the underlying (dis-) co-ordination to be due to the complex dynamic interactions between VSCMs and ECM.

## Underlying extracellular matrix and cellular remodelling

In alignment with prior studies we found a reduction in the fraction of elastin accompanied by impaired quality of elastin fibres in ATAA (Choudhury et al., 2009; Nightingale et al., 2022; Yousef et al., 2021) and ATAAD (Wang et al., 2005) patients. Particularly in ATAAD elastin degradation was perspicuous, even when compared to ATAAs. Elastin degradation is a hallmark of arterial stiffening and aortic aneurysm formation, and it affects arterial compliance function (Parikh et al., 2021). Interestingly elastin degradation was associated with increased proteolytic activity of MMP-2 and MMP-9. The deterioration of elastin fibres in the arterial wall is

pathophysiologically relevant because the production of functional elastin is completed during postnatal development (Kelleher et al., 2004), and the resulting aortic stiffness is not compensated by renewed elastin formation.

Although elastin fragmentation and degradation have been linked to ATAAD, the role of collagen remodelling is still not well defined. Despite being a principal component of the ECM the involvement of collagen in aortic pathology remains unclear, with numerous conflicting reports (Jana et al., 2019; Mimler et al., 2019). Both increased (Wang et al., 2006; Whittle et al., 1990) and decreased collagen (de Figueiredo Borges et al., 2008) content have been linked to weakening the aortic wall, leading to aneurysm formation and, in some cases, dissection (Sariola et al., 1986). Our findings corroborate reduced collagen content in some of our ATAA cases, indicating that not all ATAAs exhibit the same characteristics. This considerable variation in

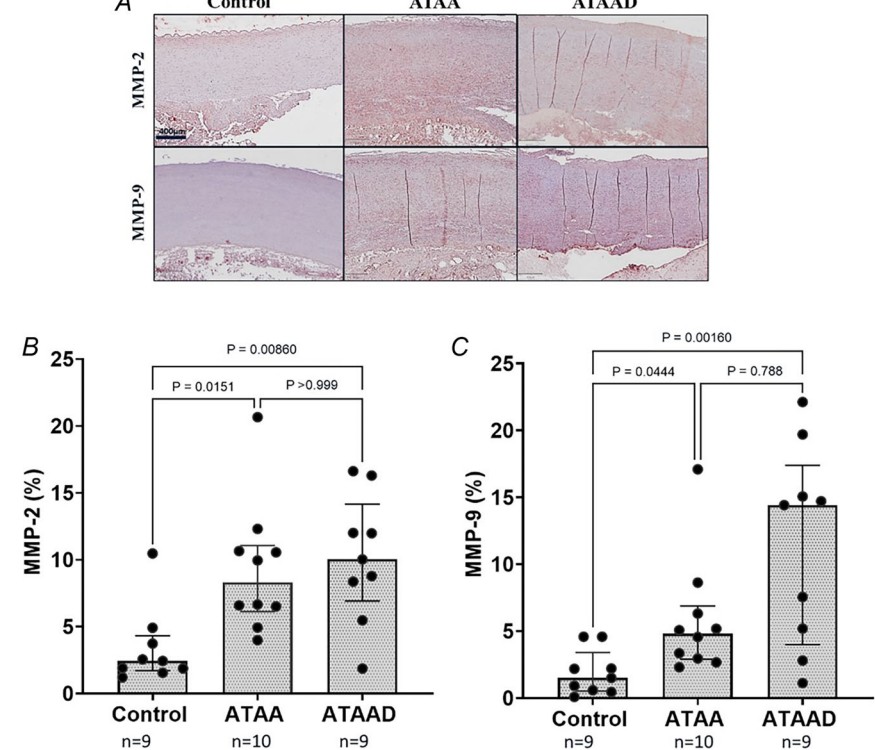

**Figure 12. Increased levels of matrix metalloproteinases (MMPs) were observed in ascending thoracic aortic aneurysm (dissection) ATAA(D) in the anterior aorta**

*A*, immunohistochemical (IHC) staining of transverse sections of the ascending aorta from control, ATAA and ATAAD patients, visualizing the proteolytic profile of MMP-2 and MMP-9 at $10\times$ magnification. Note: Due to the optimization of antibodies for MMP-2 and MMP-9 in thoracic aortic tissue a limited number of samples were analysed: controls (9/12), ATAA (10/57) and ATAAD (9/17). Statistical tests were performed using Kruskal–Wallis test. Relative expression of MMP-2 and MMP-9 was significantly higher in both ATAA ($P = 0.0151$ and $P = 0.0444$, respectively) and ATAAD ($P = 0.0086$ and $P = 0.0016$, respectively) compared to controls, with no significant differences between ATAA and ATAAD (MMP-2: $P > 0.999$; MP-9: $P = 0.788$). Median [25th percentile to 75th percentile] values for MMP-2: control: 2.44[1.72-4.33]; ATAA: 8.31[6.12–11.07]; and ATAAD: 10.03[6.92–14.16]; and for MMP-9: control: 1.51[0.53–3.41]; ATAA: 4.83[2.89–6.90]; and ATAAD: 14.43[4.01–17.39]. [Colour figure can be viewed at wileyonlinelibrary.com]

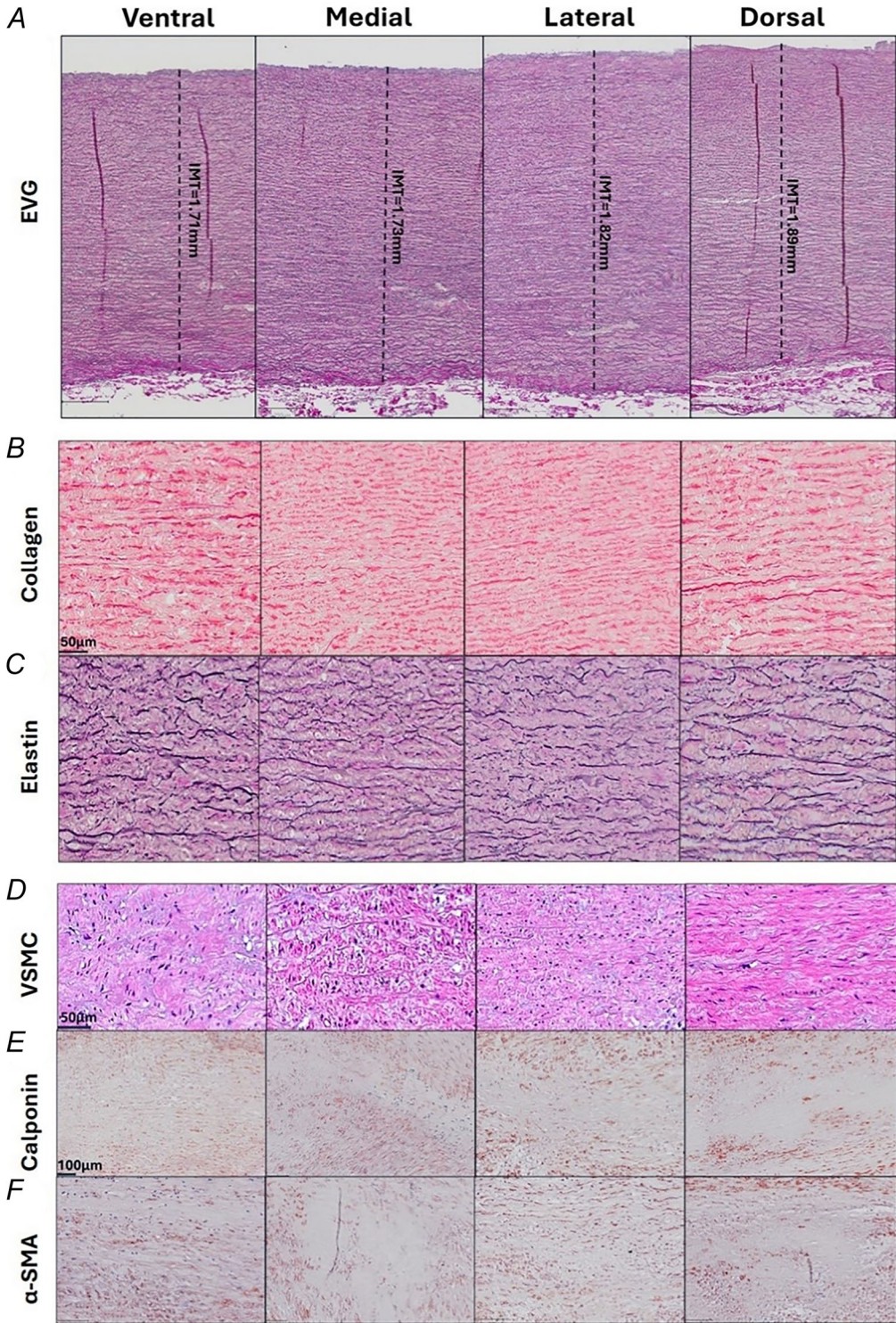

**Figure 13. Representative cases of aortic ring histology demonstrating regional uniformity in structural and cellular features**

Full-circumference aortic ring histology illustrating a case with no interregional differences in (*A*) intima-media thickness (IMT), (*B*) relative collagen percentage, (*C*) relative elastin percentage, (*D*) vascular smooth muscle cell (VSMC) density, (*E*) relative calponin expression and (*F*) relative α-smooth muscle actin (α-SMA) expression between four anatomical regions: anterior/ventral (AV), inner curvature/medial (AM), outer curvature/lateral (AL) and posterior/dorsal (AD). [Colour figure can be viewed at wileyonlinelibrary.com]

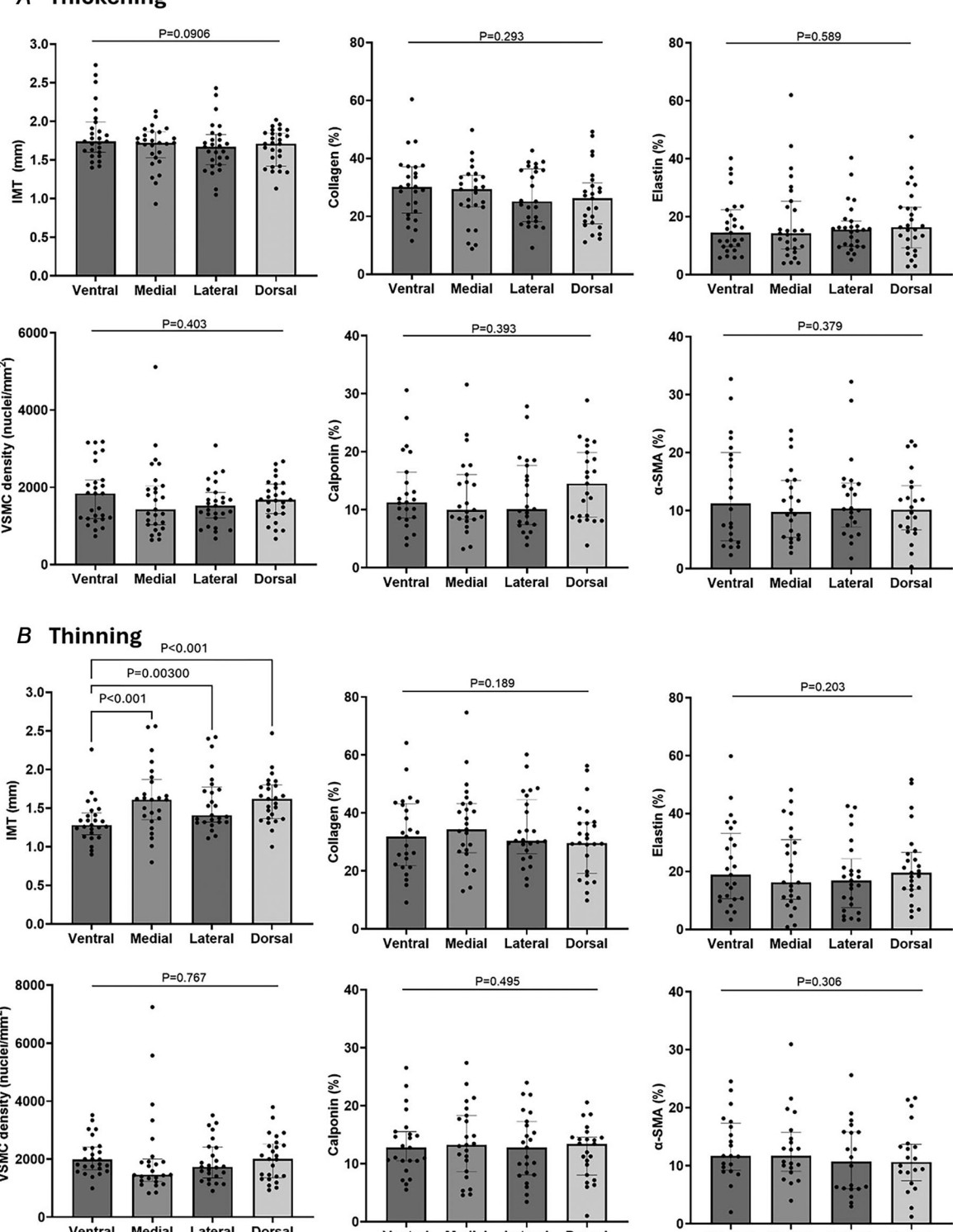

**Figure 14. Intrapatient analysis to identify differences in ascending thoracic aortic aneurysm (ATAA) wall properties across the full circumference, stratified based on the thickening-thinning dichotomy (as defined in Fig. 3)**
Note: full circumference segments were available in 52 of 57 cases ($n = 52/57$). Statistical analysis was performed using repeated-measures ANOVA (paired ANOVA) followed by Tukey's *post hoc* multiple comparisons test. In the thickening ATAA subgroup ($n = 27$) there were no differences across the circumference for intima-media thickness

(IMT). This contrasts with the thinning subgroup ($n = 25$), where we did observe ventral IMT to be significantly lower than in the other three segments. Lateral IMT, that is, in the outer bend, also tended to be lower than dorsal and medial IMT (see plots) but did not achieve statistical significance. For collagen and elastin content, VSCM density, expression of calponin and $\alpha$-smooth muscle actin ($\alpha$-SMA) there were no statistical differences across the full circumference (of the max-dilated part of the aneurysm). For mean $\pm$ SD values refer to Table 3.

**Table 3. Mean $\pm$ SD values for the full aortic circumference corresponding to Fig. 14, shown for thickening and thinning variants across aortic segments**

| Thickening | AV (mean $\pm$ SD) | AM (mean $\pm$ SD) | AL (mean $\pm$ SD) | AD (mean $\pm$ SD) |
|---|---|---|---|---|
| IMT (mm) | 1.84 $\pm$ 0.35 | 1.67 $\pm$ 0.27 | 1.67 $\pm$ 0.32 | 1.67 $\pm$ 0.23 |
| Collagen% | 30.09 $\pm$ 11.01 | 27.62 $\pm$ 10.18 | 27.01 $\pm$ 9.58 | 26.02 $\pm$ 10.77 |
| Elastin% | 16.72 $\pm$ 9.77 | 18.23 $\pm$ 13.84 | 15.99 $\pm$ 8.19 | 18.19 $\pm$ 11.04 |
| VSMC density (1/mm$^2$) | 1811 $\pm$ 770 | 1692 $\pm$ 962 | 1557 $\pm$ 559 | 1672 $\pm$ 530 |
| Calponin% | 12.99 $\pm$ 6.68 | 12.34 $\pm$ 6.71 | 12.42 $\pm$ 6.47 | 14.60 $\pm$ 6.34 |
| $\alpha$-SMA% | 13.11 $\pm$ 8.95 | 10.93 $\pm$ 6.18 | 12.15 $\pm$ 7.31 | 10.77 $\pm$ 5.90 |
| Thinning | | | | |
| IMT (mm) | 1.33 $\pm$ 0.28 | 1.62 $\pm$ 0.44 | 1.58 $\pm$ 0.38 | 1.61 $\pm$ 0.31 |
| Collagen% | 32.24 $\pm$ 13.10 | 35.41 $\pm$ 13.92 | 34.04 $\pm$ 11.81 | 30.59 $\pm$ 12.23 |
| Elastin% | 22.05 $\pm$ 14.32 | 20.84 $\pm$ 13.71 | 18.17 $\pm$ 12.18 | 21.78 $\pm$ 12.71 |
| VSMC density (1/mm$^2$) | 2101 $\pm$ 641 | 2069 $\pm$ 1508 | 1921 $\pm$ 731 | 2029 $\pm$ 775 |
| Calponin% | 13.32 $\pm$ 5.39 | 13.82 $\pm$ 6.28 | 12.92 $\pm$ 5.84 | 12.08 $\pm$ 4.68 |
| $\alpha$-SMA% | 13.20 $\pm$ 5.64 | 13.05 $\pm$ 6.20 | 11.13 $\pm$ 6.21 | 11.33 $\pm$ 5.77 |

Abbreviations: IMT, intima-media thickness; $\alpha$-SMA, $\alpha$-smooth muscle actin; VSMC, vascular smooth muscle cell.

collagen metabolism among ATAAs may provide crucial insights into the factors determining the development of ATAAD in some patients while others remain unaffected. Disruptions to collagen metabolism, expression and arrangement may compromise aortic wall integrity and lead to ATAAD. Unlike elastin collagen regeneration in ECM occurs throughout its lifetime (Nissen et al., 1978; Sluijter et al., 2004), preserving structural strength for short-lived mechanical challenges. The range of differences in collagen content in ATAA patients could explain the discrepancy between aneurysm formation and the risk of dissection or rupture.

Contrary to the hypothesis that VSMC death or loss drives aneurysm formation (Schmid et al., 2003) and vascular ageing (Waters et al., 2017) our observations in ATAA cases revealed (mal)adaptive changes in VSMCs. First we observed a local increase in VSMC density in both ATAA and ATAAD accompanied by a reduction in the expression of contractile phenotype markers, specifically calponin and $\alpha$-SMA. This indicates a transition of VSMCs from the contractile phenotype towards a synthetic (proliferative, migratory) phenotype. Similar findings were reported by Yousef et al. (Yousef et al., 2021) and Kirsh et al. (Kirsch et al., 2006), demonstrating a hyperplasia of VSMCs in ATAA, which may indicate a compensatory process for elastin loss. Second we observed a positive correlation between VSMC density and inter-lamellar distance. It is worth noting that increased VSMC density may reflect a compensatory mechanism where VSMCs change phenotype and become synthetic to contribute to the vessel's elasticity and help maintain and repair the overall vascular structure in response to injury or stress (Martin-Blazquez et al., 2021). Notably it seems that this local increase in VSMC density is not associated with an inflammatory response, further suggesting a structural rather than immunological role in vessel remodelling (Leone et al., 2020; Stella et al., 2022).

Regardless of a primary or secondary phenomenon in sporadic aortic aneurysms we hypothesize that the phenotypic switch of VSMCs increases the expression of MMP-2 and MMP-9 in ATAA and ATAAD, thereby aiding the degradation of elastin and collagen (Martin-Blazquez et al., 2021). Furthermore we found even greater expression of MMP-2 and MMP-9 in ATAAD. Similar findings were reported earlier (Koullias et al., 2004), suggesting that MMP-2 and MMP-9 have an important significance in the development and progression of ATAAD. Furthermore this transformation into a proteolytic state alters mechanotransduction, which exaggerates the maladaptive remodelling of the vessel wall (Humphrey et al., 2014).

## Translational perspective

The interpatient variability we observed at the cell-matrix level may explain why ATAA dimensions do not show a one-to-one relation with the occurrence of dissection in clinical practice (Adriaans et al., 2021; Ganizada et al., 2024; Perez et al., 2023; Rylski et al., 2014).

The present study underpins the crucial need for improved prognostic markers (Ganizada et al., 2024) to distinguish between ATAA patients vulnerable to developing dissection/rupture and ATAA with adaptive (stable) remodelling. If it becomes clinically possible to reliably monitor the diameter-IMT-ratio in patients, aneurysm development could be assessed in terms of wall thinning or thickening, that is, reflecting greater or lesser loss of wall stress homeostasis. However routine medical imaging currently fails to measure aortic wall thickness accurately due to limited resolution (Debeij et al., 2023). Recent advances in CT imaging seem promising (Hagar et al., 2024; Rotzinger et al., 2020), potentially enabling clinical monitoring of wall thickness changes over time and along the circumference.

Considering the crucial role of VSMCs in maintaining tissue homeostasis our study further supports the development of cell-based treatment strategies aimed at preventing or delaying life-threatening aortopathies and their associated events.

### Limitations of our study

One limitation of our study is the comparison between *in vivo* diameter and *ex vivo*/histological IMT measurements. This may lead to misinterpretation of their correlation and values of the diameter-to-(IMT×2) ratio due to differences in blood pressure conditions and axial stretch (Parikh et al., 2024). *In vivo* measurements are subject to haemodynamic forces, including blood pressure fluctuations, which influence vessel dimensions. In contrast *ex vivo* or histological measurements are taken under static conditions, where the influence of haemodynamic forces is absent. This discrepancy in measurement conditions can lead to discrepancies in the observed relationship between diameter and IMT. Additionally the diameter-to-(IMT×2) ratio, often used as a metric in vascular studies (Semmler et al., 2021), may be influenced by these differences in haemodynamic load, potentially leading to erroneous conclusions regarding arterial remodelling or pathology.

We did not evaluate pulse wave velocity (PWV) in our study population, for instance using 4D-flow MR imaging (MRI) locally or by carotid-femoral applanation tonometry as a proxy, therefore we cannot correlate our findings to age-related stiffening that might contribute to medial remodelling/arteriosclerosis in our population.

We did not fully evaluate the multiple and complex associations between vessel, cell and matrix properties. Our present analysis strongly supports such an approach. However to conduct such statistical analyses in a well-powered manner the current population sample size is simply insufficient. Our platform approach is underway to achieve more than $n = 300$ cases (Ganizada et al., 2023), allowing for robust further evaluation.

### Conclusion

The present study reveals that – during ATAA formation – the homoeostatic regulation of wall stress may remain partially intact, particularly in 'thickening' but clearly less in 'thinning' aneurysms. Our finding supports the current critique around aortic dimensions as a prognostic marker for preventive surgical intervention.

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

## Additional information

### Data availability statement

The data of this study are available from the corresponding author upon reasonable request and in line with local and national GDPR regulations.

### Competing interests

L.J.S. has received institutional grants from Gnosis by Lesaffre and Bayer and is a shareholder of Coagulation Profile. All other authors have nothing to declare.

### Author contributions

B.H.G. designed the study, curated the data, performed formal analysis and prepared the original draft. K.D.R., E.B. and L.J.S. reviewed and edited the manuscript. A.I., M.Q.E.K. and C.F.M.M. contributed to the formal analysis. M.J.F.G.R. was responsible for visualization. J.P.C., B.S. and K.L. developed the software. A.M.J. performed validation. J.G.M. and L.J.S. provided material resources. S.P., P.J.M.H.S., R.R.K. and R.L. contributed to data collection and analysis and critically reviewed the manuscript. L.J.S., E.B. and K.D.R. supervised the study. All authors commented on the manuscript and approved its submission for publication.

### Funding

European Union's Horizon 2020 research and innovation programme under the Marie Sklodowska-Curie: Leon Schurgers, 675111; European Union's Horizon 2020 research and innovation programme under the Marie Slodowska-Curie: Leon Schurgers, 722609.

### Acknowledgements

We extend our gratitude to the following funding organizations for their financial support. This research was partially financially supported through a PPP Allowance made available by Health-Holland, Top-Sector Life Sciences & Health, to the Association of Collaborating Health Foundations (SGF) to stimulate public-private partnerships, and by ZonMW, grant LSHM21078-SGF 'CELLSYSTEMICS'.

## Keywords

matrix remodelling, mechanical homeostasis, mechanosensing, vascular smooth muscle, wall stress

## Supporting information

Additional supporting information can be found online in the Supporting Information section at the end of the HTML view of the article. Supporting information files available:

**Peer Review History**

