## [Peer Review History · The Journal of Physiology]

Loss of Wall Stress Homeostasis in Ascending Thoracic Aortic Aneurysm: Histomorphometric Insights into Patient Variants

Berta Ganizada, Koen D Reesink, Shaiv Parikh, Pepijn Saraber, Jack Cleutjens, Austin Isabella, Mathilde Krebbekx, Armand Jaminon, Cecile Maassen, Mitch Ramaekers, Bart Spronck, Koen W.F. van der Laan, Rory Koenen, Roberto Lorusso, Jos Maessen, Elham Bidar, and Leon Schurgers

DOI: 10.1113/JP288734

Corresponding author(s): Berta Ganizada (berta.ganizada@maastrichtuniversity.nl)

The following individual(s) involved in review of this submission have agreed to reveal their identity: Anton Kutikhin (Referee #2)

Review Timeline:	Submission Date:	13-Feb-2025
	Editorial Decision:	22-Apr-2025
	Revision Received:	16-May-2025
	Accepted:	10-Jun-2025

Senior Editor: Harold Schultz

Reviewing Editor: Nikki Jernigan

Transaction Report:

Dear Dr Ganizada,

Re: JP-RP-2025-288734 "Loss of Wall Stress Homeostasis in Ascending Thoracic Aortic Aneurysm: Histomorphometric Insights into Patient Variants" by Berta Ganizada, Koen D Reesink, Shaiv Parikh, Pepijn Saraber, Jack Cleutjens, Austin Isabella, Mathilde Krebbekx, Armand Jaminon, Cecile Maassen, Mitch Ramaekers, Bart Spronck, Koen W.F. van der Laan, Rory Koenen, Roberto Lorusso, Jos Maessen, Elham Bidar, and Leon Schurgers

Thank you for submitting your manuscript to The Journal of Physiology. It has been assessed by a Reviewing Editor and by 2 expert referees and we are pleased to tell you that it is acceptable for publication following satisfactory revision.

REVISION CHECKLIST:

We look forward to receiving your revised submission.

Yours sincerely,

Harold Schultz
Senior Editor
The Journal of Physiology

EDITOR COMMENTS

Reviewing Editor:

Comments for Authors to ensure the paper complies with the Statistics Policy:

The exact p values must be stated to three significant figures (not decimal places) even when 'no statistical significance' is being reported. For most figures, these can be put on the figure instead of asterisks.

Comments to the Author:

Nicely conducted study and of clinical relevance, please consider adding a Translational Perspective paragraph

Use exact p-values even when 'no statistical significance' is being reported. For most figures, these can be put on the figure instead of asterisks.

Senior Editor:

Comments for Authors to ensure the paper complies with the Statistics Policy:

Data summaries must be defined as mean with standard deviation (SD) (standard error is not acceptable). Variance is never defined in the manuscript that we could find. Please make SD clear in the data analysis section or in table/figure legends.

Provide the magnitude of the test statistic (p value) in the figures, figure legend, or explicitly cited in the main text when referring to the figure. The exact p values must be stated to three significant figures even when 'no statistical significance' is being reported. As an exception, stating $p < 0.001$ is permitted. Please include actual p values as stated here in Fig. 2, 4-11, S3-S4, and Table 2. Symbols are not permitted.

Please state the statistical test(s) used in tables and figure legends when statistics are reported.

Clearly state all relevant 'n' values in the main text, figures, and tables. State sample size for each experimental group and whether numbers are replications of the same experimental sample, multiple samples taken from the same source, independent samples taken from a population, or a combination of these replicates (e.g., x replications in y animals). Indicate when samples were excluded for any reason.

Comments to the Author:

Thank you for submitting your research article for consideration in the Journal of Physiology. The article has been reviewed by experts in the field and found to be potentially acceptable for publication pending adequate revision to address all of the concerns raised. Please address all comments from the external referees and reviewing editor as well as address the list of requirements for publication in the journal including the following.

1. The following sections are required: Data availability statement. Figure legends must be listed in a section after Additional Information.

2. Please include an Abstract figure and legend, and first author profile

3. The Journal does not permit data as supplements for transparency in PDFs to readers. Please incorporate Figs S1- S4 into the results as regular figures.

4. The statistical reporting does not meet Journal requirements:

Data summaries must be defined as mean with standard deviation (SD) (standard error is not acceptable). Variance is never defined in the manuscript that we could find. Please make SD clear in the data analysis section or in table/figure legends.

Provide the magnitude of the test statistic (p-value) in the figures, figure legend, or explicitly cited in the main text when referring to the figure. The exact p values must be stated to three significant figures even when 'no statistical significance' is being reported. As an exception, stating $p < 0.001$ is permitted. Please include actual p-values as stated here in Fig. 2, 4-11, S3-S4, and Table 2. Symbols are not permitted.

Please state the statistical test(s) used in tables and figure legends when statistics are reported.

Clearly state all relevant 'n' values in the main text, figures, and tables. State sample size for each experimental group and whether numbers are replications of the same experimental sample, multiple samples taken from the same source, independent samples taken from a population, or a combination of these replicates (e.g., x replications in y animals). Indicate when samples were excluded for any reason.

REFeree COMMENTS

Referee #1:

The work presented by Ganizada and collaborators primarily compares the phenotypic changes and elastin thickness between ATAA and ATAAD and highlights the heterogeneity of analyzing only one region or correcting data based on a single region when evaluating aortic diseases. The authors used standardized methods and also performed a double-check to exclude possible bias. In general, the study is interesting, the text is enjoyable to read, and it has a meaningful rationale with an important inquiry in the literature. In general the study is interesting, the text is enjoyable to read, the limitations are well discussed along the text and it has a meaningful rationale with an important inquiry in the literature. I have minor suggestions.

Minor

Very interesting results, considering patients age and arteriosclerosis pathophysiology did the group evaluated patients Pulse Wave Velocity (PWV) between ATAA more thickening and thinning groups separated in figure 3? is there a difference?

Paragraph 117 - Method 2.3 starts with "or histological examination" is it a phrase missing? Please check.

Figure 9 and 10 - could the authors specify in the legends the statistical test chosen to compare the groups in this 2 figures?

Paragraph 350 - Phrase "A previous study, in which we found all stress..." could it be meant "wall stress"?

Referee #2:

Here, Ganizada and colleagues investigated histomorphometric insights into the development of ascending thoracic aortic aneurysms (including dissections).

The identification of distinct association patterns between aortic radius (AR) and intima-media thickness (IMT) is novel and might be highly valuable in clinical practice; authors showed that control patients with ascending thoracic aortic aneurysm (ATAA) with a lower AR/IMT ratio tend to have higher aortic wall thickening in comparison with patients with ascending thoracic aortic aneurysm dissection (ATAAD) who tend to have a higher AR/IMT ratio. Importantly, authors also showed significantly lower collagen content, elastic fraction, and lamella thickness (and also higher interlamellar distance and number of disruptions) in patients with ATAA and ATAAD (which is evident) but additionally demonstrated lower collagen content, reduced lamella thickness, and increased interlamellar distance in patients with ATAAD.

Analysis of vascular smooth muscle cell content indicated an increased density, altered appearance of VSMCs, and reduced alpha smooth muscle cell and calponin content in both ATAA and particularly ATAAD. Also having considered a correlation between VSMC density and interlamellar distance and increased production of MMP-2 and MMP-9, authors insightfully suggested a compensatory phenotypic switch of VSMCs in response to the degradation of elastin. The authors concluded on two phenotypes of aortic wall response to aneurysmal development (thickening and thinning morphotypes).

The paper has a good concept, outstanding technical quality, and is written in an intelligible manner. The study design as well as materials and methods have no glaring omissions. This manuscript is indeed influential because it critically summarises the clinical phenotypes of intact and dissected ascending thoracic aortic aneurysms.

END OF COMMENTS

Response to Reviewer and Editor Comments

We thank the editors and reviewers for their thoughtful and constructive comments.

We have carefully considered all feedback and revised the manuscript accordingly.

Below, we provide a point-by-point response to each comment, and revisions to the main text are quoted *in italics*.

Textual changes are indicated **in the highlights** of the revised manuscript.

To Reviewing Editor:

Comment 1: Comments for Authors to ensure the paper complies with the Statistics Policy: The exact p values must be stated to three significant figures (not decimal places) even when 'no statistical significance' is being reported. For most figures, these can be put on the figure instead of asterisks.

Response: We acknowledge that our original submission was not yet in line with the Journal's statistics policy. We have revised the manuscript to report exact p-values to three significant figures throughout; we have replaced asterisks in all figures with the corresponding exact p-values; all figures, legends, and main text have been revised and updated.

Comment 2. Nicely conducted study and of clinical relevance, please consider adding a Translational Perspective paragraph

Response: In response to the suggestion, we have added a "Translational Perspective" paragraph, now reading (section 4.3, page 30):

4.3 *Translational perspective*

The inter-patient variability at the cell-matrix level we found may explain why ATAA dimensions do not show a one-to-one relation with the occurrence of dissection in clinical practice (Rylski et al., 2014; Adriaans et al., 2021; Perez et al., 2023; Ganizada et al., 2024). The present study underpins the crucial need for improved prognostic markers (Ganizada et al., 2024) to distinguish between ATAA patients vulnerable to developing dissection/rupture and ATAA with adaptive (stable) remodelling. If it becomes clinically possible to reliably monitor the diameter-IMT-ratio in patients, aneurysm

development could be assessed in terms of wall thinning or thickening, i.e., reflecting greater or lesser loss of wall stress homeostasis. However, routine medical imaging currently fails to measure aortic wall thickness accurately due to limited resolution (Debeij et al., 2023). Recent advances in CT-imaging seem promising (Rotzinger et al., 2020; Hagar et al., 2024), potentially enabling clinical monitoring of wall thickness changes over time and along the circumference.

Considering the crucial role of (vascular smooth muscle) cells in maintaining tissue homeostasis, our study further supports the development of cell-based treatment strategies aimed at preventing or delaying life-threatening aortopathies and their associated events.

Comment 3: Use exact p-values even when 'no statistical significance' is being reported. For most figures, these can be put on the figure instead of asterisks.

Response: Adjusted, please refer to our response to comment 1.

To Senior Editor:

Comment 1: Comments for Authors to ensure the paper complies with the Statistics Policy: Data summaries must be defined as mean with standard deviation (SD) (standard error is not acceptable). Variance is never defined in the manuscript that we could find. Please make SD clear in the data analysis section or in table/figure legends.

Response: We have revised the statistical analysis section 2.6 to clearly describe how data distribution was handled in our descriptive and testing statistical methods. Accordingly, we have added data distributions (normally distributed data is reported as mean \pm SD and not normally distributed data as median with IQR) and statistical tests explicitly in all figure legends. Refer to Table 3 (page 27) for the mean \pm SD values corresponding to the data presented in Figure 14.

Section 2.6, page 10 now reading:

2.6 Statistical analysis

Statistical analysis was performed using GraphPad Prism 10 (GraphPad Software, San Diego, USA).

For continuous variables, normality was assessed using the Shapiro-Wilk test, indicated by P-values > 0.05 . All normally distributed data is presented as mean \pm standard deviation (SD) along with corresponding n values. Skewed data (Shapiro-Wilk P-value ≤ 0.05) is presented as median with interquartile ranges [25th percentile-75th percentile].

Inter-group comparisons for skewed data were performed using Kruskal-Wallis testing, followed by Dunn's test for pairwise comparisons. Inter-group comparisons for normal data were performed using one-way analysis of variance (ANOVA), followed by Tukey's multiple comparison tests. For inter-group comparisons (Control, ATAA, ATAAD), each data point in the swarm and correlation plots represents an independent biological sample from a single patient (i.e., n equals the number of patients per group).

For intra-group comparisons among the four anatomical regions within the ATAA group (see Figure 1), repeated measures ANOVA with Geisser-Greenhouse correction was used for equal variability of differences. Post-hoc pairwise comparisons were conducted using Tukey's multiple comparisons test.

For intra-group comparisons, each patient represents four circumferential measurements (AV, AM, AL, AD), which are replicates from the same tissue sample.

Chi-squared tests were used to compare categorical variables between groups, while Spearman correlation was applied to assess relationships between quantitative histological data. All statistical tests were two-tailed, and a P-value of <0.05 was considered statistically significant.

Comment 2: Provide the magnitude of the test statistic (p value) in the figures, figure legend, or explicitly cited in the main text when referring to the figure. The exact p values must be stated to three significant figures even when 'no statistical significance' is being reported. As an exception, stating $p < 0.001$ is permitted). Please include actual p values as stated here in Fig. 2, 4-11, S3-S4, and Table 2. Symbols are not permitted.

Response: In line with our response to the reviewing editor (see above comments 1 and 3, page 1 and 2), we revised our manuscript accordingly. All asterisk symbols or other symbolic notations previously used to indicate significance have been removed and replaced with the corresponding exact p-values, either directly on the figures, legends, tables, and main text.

Comment 3: Please state the statistical test(s) used in tables and figure legends when statistics are reported.

Response: Please refer to our response to Comment 1. As noted there, we have revised the Statistical Analysis section (Section 2.6, page 10) to more clearly describe the statistical methods used. In addition, we also added that statistical tests in all figure and table legends

Comment 4: Clearly state all relevant 'n' values in the main text, figures, and tables. State sample size for each experimental group and whether numbers are replications of the same experimental sample, multiple samples taken from the same source, independent samples taken from a population, or a combination of these replicates (e.g., x replications in y animals). Indicate when samples were excluded for any reason.

Response: We thank the senior editor for these important comments.

Sample sizes for each group are now specified in the main text, figure, and tables.

In the “statistical analysis section 2.6, page 10”, we now explicitly clarify whether the number of replicants of each experiment is from the same source or independent samples, now reading:

“Inter-group comparisons (Control, ATAA, ATAAD), each data point in the swarm and correlation plots represents an independent biological sample from a single patient (i.e., n equals the number of patients per group).”

“Intra-group comparisons, each patient represents four circumferential measurements (AV, AM, AL, AD), which are replicates from the same tissue sample.”

Sample exclusions, if any, are indicated in the respective figure legends.

Comment 5: The following sections are required: Data availability statement. Figure legends must be listed in a section after Additional Information.

Response: Thank you for pointing this out.

We have added a Data Availability Statement in the revised manuscript, see page 33. Herein, we state that “the data of this study are available from the corresponding author upon reasonable request and in line with local and national GDPR regulations”.

All figure legends have been moved after the Additional Information section, as required.

Comment 6: Please include an Abstract figure and legend, and the first author's profile

Response: We have added an abstract figure (descriptive figure of the main findings), see page 3 of the manuscript.

We have also included a First Author Profile as a separate Word document.

Comment 7: The Journal does not permit data as supplements for transparency in PDFs to readers. Please incorporate Figs S1- S4 into the results as regular figures.

Response: We have incorporated Figures S1–S4 into the main manuscript as regular figures and referenced them appropriately within the results section. These figures are now renumbered accordingly in the revised version.

Comment 8: Data summaries must be defined as mean with standard deviation (SD) (standard error is not acceptable). Variance is never defined in the manuscript that we could find. Please make SD clear in the data analysis section or in table/figure legends.

Response: Please refer to our response to Comment 1. As noted there, we have revised the Statistical Analysis section (Section 2.6, page 10) to more clearly describe the statistical methods used. In addition, we also added data distributions in all figure legends (normally distributed data is reported as mean \pm SD and not normally distributed data as median with IQR).

Comment 9: Provide the magnitude of the test statistic (p-value) in the figures, figure legend, or explicitly cited in the main text when referring to the figure. The exact p values must be stated to three significant figures even when 'no statistical significance' is being reported. As an exception, stating $p < 0.001$ is permitted. Please include actual p-values as stated here in Fig. 2, 4-11, S3-S4, and Table 2. Symbols are not permitted.

Response: In line with our response to the reviewing editor (see above comments 1 and 3), we revised our manuscript accordingly. All asterisk symbols or other symbolic notations previously used to indicate significance have been removed and replaced with the corresponding exact p-values, either directly on the figures, legends, tables, and main text

Comment 10: Please state the statistical test(s) used in tables and figure legends when statistics are reported.

Response: We have revised all tables and figure legends to clearly state the statistical tests used for the reported analyses.

Comment 11: Clearly state all relevant 'n' values in the main text, figures, and tables. State sample size for each experimental group and whether numbers are replications of the same experimental sample, multiple samples taken from the same source, independent samples taken from a population, or a combination of these replicates (e.g., x replications in y animals). Indicate when samples were excluded for any reason.

Response: Adjusted, please refer to our response to Comment 4.

Referee Comments

Referee #1:

Comment 1: Very interesting results, considering patients age and arteriosclerosis pathophysiology did the group evaluated patients Pulse Wave Velocity (PWV) between ATAA more thickening and thinning groups separated in figure 3? is there a difference?

Response: Thank you for the suggestion. Unfortunately, we did not evaluate PWV in our study population (e.g., by 4D-flow MRI locally or by carotid-femoral applanation tonometry as a proxy). We do discuss this now in our limitations section, reading (Section 4.4, page 31):

We did not evaluate PWV in our study population, for instance by 4D-flow MRI locally or by carotid-femoral applanation tonometry as a proxy, therefore, we cannot correlate our findings to age-related stiffening that might contribute to medial remodelling/arteriosclerosis in our population.

Comment 2: Paragraph 117 - Method 2.3 starts with "or histological examination" is it a phrase missing? Please check.

Response: Indeed, in section 2.3, this is a typographical error. We corrected it accordingly. now reading (Section 2.3, page 9):

For histological examination, all tissues were fixed in 1% HEPES buffered formalin solution, adjusted to pH 7.4 at room temperature for 24 hours, and next embedded in paraffin.

Comment 3: Figure 9 and 10 - could the authors specify in the legends the statistical test choosen to compare the groups in this 2 figures?

Response: We have added statistical tests to the figure legends. Since the data were not normally distributed, we used the Kruskal-Wallis test in both analyses to compare the three independent groups. Note: Figure 10 is now Figure 12.

Comment 4: Paragraph 350 - Phrase "A previous study, in which we found all stress..." could it be meant "wall stress"?

Response: Indeed, in section 4.1, this is a typographical error. We corrected it accordingly. Section 4.1, page 28 now reading:

A previous study, in which we found wall stress normalization and homogenization to be essential in capturing age-related arterial stiffening (Reesink & Spronck, 2019) corroborates the lack of circumferential differences in our ATAA cases: even during the

pathological dilatation leading to an aneurysm, wall thickness and cell-matrix properties remain homogeneous (Figure 14).

Referee #2:

Comment 1:

Here, Ganizada and colleagues investigated histomorphometric insights into the development of ascending thoracic aortic aneurysms (including dissections).

The identification of distinct association patterns between aortic radius (AR) and intima-media thickness (IMT) is novel and might be highly valuable in clinical practice; authors showed that control patients with ascending thoracic aortic aneurysm (ATAA) with a lower AR/IMT ratio tend to have higher aortic wall thickening in comparison with patients with ascending thoracic aortic aneurysm dissection (ATAAD) who tend to have a higher AR/IMT ratio. Importantly, authors also showed significantly lower collagen content, elastic fraction, and lamella thickness (and also higher interlamellar distance and number of disruptions) in patients with ATAA and ATAAD (which is evident) but additionally demonstrated lower collagen content, reduced lamella thickness, and increased interlamellar distance in patients with ATAAD.

Analysis of vascular smooth muscle cell content indicated an increased density, altered appearance of VSMCs, and reduced alpha smooth muscle cell and calponin content in both ATAA and particularly ATAAD. Also having considered a correlation between VSMC density and interlamellar distance and increased production of MMP-2 and MMP-9, authors insightfully suggested a compensatory phenotypic switch of VSMCs in response to the degradation of elastin. The authors concluded on two phenotypes of aortic wall response to aneurysmal development (thickening and thinning morphotypes).

The paper has a good concept, outstanding technical quality, and is written in an intelligible manner. The study design as well as materials and methods have no glaring omissions. This manuscript is indeed influential because it critically summarises the clinical phenotypes of intact and dissected ascending thoracic aortic aneurysms.

Response: We sincerely thank Reviewer #2 for thoughtful and encouraging comments. We are grateful for the recognition of the novelty and potential clinical relevance of our findings, particularly regarding the distinct AR/IMT ratio patterns and phenotypic remodeling of the aortic wall. We also appreciate your positive feedback on the technical quality, clarity of writing, and overall design of the study. Your comments affirm the value of our work and have been truly motivating during the revision process.

Dear Miss Ganizada,

Re: JP-RP-2025-288734R1 "Loss of Wall Stress Homeostasis in Ascending Thoracic Aortic Aneurysm: Histomorphometric Insights into Patient Variants" by Berta Ganizada, Koen D Reesink, Shaiv Parikh, Pepijn Saraber, Jack Cleutjens, Austin Isabella, Mathilde Krebbekx, Armand Jaminon, Cecile Maassen, Mitch Ramaekers, Bart Spronck, Koen W.F. van der Laan, Rory Koenen, Roberto Lorusso, Jos Maessen, Elham Bidar, and Leon Schurgers

We are pleased to tell you that your paper has been accepted for publication in The Journal of Physiology.

Yours sincerely,

Harold Schultz
Senior Editor
The Journal of Physiology

If you would like to receive our 'Research Roundup', a monthly newsletter highlighting the cutting-edge research published in The Physiological Society's family of journals (The Journal of Physiology, Experimental Physiology, Physiological Reports, The Journal of Nutritional Physiology and The Journal of Precision Medicine: Health and Disease), please click this link, fill in your name and email address and select 'Research Roundup':
<https://www.physoc.org/journals-and-media/membernews>

- You can help your research get the attention it deserves! Check out Wiley's free Promotion Guide for best-practice recommendations for promoting your work at: www.wileyauthors.com/eeo/guide. You can learn more about Wiley Editing Services which offers professional video, design, and writing services to create shareable video abstracts, infographics, conference posters, lay summaries, and research news stories for your research at: www.wileyauthors.com/eeo/promotion.

EDITOR COMMENTS

Senior Editor:

Comments to the Author:

The editors thank the authors for these final adjustments to the manuscript. The article is now accepted for publication. Congratulations for an interesting and insightful study. Please consider the Journal of Physiology for your future studies.

REFEREE COMMENTS

Referee #1:

Dear authors, thank you for the changes and your revised article with the discussed suggestions in accordance with the journal's guidelines.

I have no further revisions and I consider the manuscript suitable for publication in its current form.